# Metabolic coupling between soil aerobic methanotrophs and denitrifiers in rice paddy fields

Kang-Hua Chen[1,2], Jiao Feng [1,2] ✉, Paul L. E. Bodelier[3], Ziming Yang [4], Qiaoyun Huang [1,2], Manuel Delgado-Baquerizo [5], Peng Cai [1,2], Wenfeng Tan [2] & Yu-Rong Liu [1,2] ✉

Paddy fields are hotspots of microbial denitrification, which is typically linked to the oxidation of electron donors such as methane ($CH_4$) under anoxic and hypoxic conditions. While several anaerobic methanotrophs can facilitate denitrification intracellularly, whether and how aerobic $CH_4$ oxidation couples with denitrification in hypoxic paddy fields remains virtually unknown. Here we combine a ~3300 km field study across main rice-producing areas of China and $^{13}CH_4$-DNA-stable isotope probing (SIP) experiments to investigate the role of soil aerobic $CH_4$ oxidation in supporting denitrification. Our results reveal positive relationships between $CH_4$ oxidation and denitrification activities and genes across various climatic regions. Microcosm experiments confirm that $CH_4$ and methanotroph addition promote gene expression involved in denitrification and increase nitrous oxide emissions. Moreover, $^{13}CH_4$-DNA-SIP analyses identify over 70 phylotypes harboring genes associated with denitrification and assimilating $^{13}C$, which are mostly belonged to *Rubrivivax*, *Magnetospirillum*, and *Bradyrhizobium*. Combined analyses of $^{13}C$-metagenome-assembled genomes and $^{13}C$-metabolomics highlight the importance of intermediates such as acetate, propionate and lactate, released during aerobic $CH_4$ oxidation, for the coupling of $CH_4$ oxidation with denitrification. Our work identifies key microbial taxa and pathways driving coupled aerobic $CH_4$ oxidation and denitrification, with important implications for nitrogen management and greenhouse gas regulation in agroecosystems.

Nitrogen (N) fertilization application has dramatically increased the amount of N in terrestrial ecosystems globally[1]. Rice is one of the top three main crops for N fertilizer consumption worldwide, and rice croplands receive more than 15% of global N fertilizer applications[2].

However, N fertilizer use efficiency in the paddy fields is relatively low, with a range between 20 and 40% of the applied N fertilizers[3]. Microbial denitrification is a fundamental process causing N losses to the atmosphere in paddy fields, which is often mediated by conventionally

[1]National Key Laboratory of Agricultural Microbiology and College of Resources and Environment, Huazhong Agricultural University, Wuhan 430070, China. [2]State Environmental Protection Key Laboratory of Soil Health and Green Remediation and Hubei Key Laboratory of Soil Environment and Pollution Remediation, Huazhong Agricultural University, Wuhan 430070, China. [3]Department of Microbial Ecology, Netherlands Institute of Ecology (NIOO-KNAW), PO Box 50, 6700 AB Wageningen, The Netherlands. [4]Department of Chemistry, Oakland University, Rochester, MI 48309, USA. [5]Laboratorio de Biodiversidad y Funcionamiento Ecosistémico, Instituto de Recursos Naturales y Agrobiología de Sevilla (IRNAS), CSIC, Sevilla 41012, Spain. ✉e-mail: fengjiao@mail.hzau.edu.cn; yrliu@mail.hzau.edu.cn

known denitrifiers[4,5]. Conceptual and empirical advancements suggest that soil denitrification might be linked to the oxidation of biochemically relevant electron donors, including methane ($CH_4$) produced under anoxic and hypoxic conditions[6,7]. For example, some anaerobic $CH_4$ oxidizers (e.g., the NC10 bacteria) in deep soil or sediment can oxidize $CH_4$ coupled to denitrification intracellularly with nitrite ($NO_2^-$) as the electron acceptor[8,9]. However, taxa capable of anaerobic $CH_4$ oxidation coupled to denitrification were generally not identified in the surface layer of paddy fields[10,11]. This is attributed to the structured nature of paddy fields wherein the surface layer is partially oxic or hypoxic, primarily due to the leakage and diffusion of oxygen ($O_2$) released from the rice roots[12,13]. Remarkably, it is estimated that over 70% of the produced $CH_4$ in hypoxic conditions is consumed by aerobic methanotrophs before escaping to the atmosphere[14,15]. Aerobic $CH_4$ oxidation has been found to significantly promote N removal efficiency via denitrification in biofilm reactors[16]. Paddy fields are hotspots of both microbial denitrification and aerobic $CH_4$ oxidation, and support the co-occurrence of highly abundant and diverse aerobic and anaerobic microbes including aerobic methanotrophs and denitrifiers[17,18]. However, microbial taxa and metabolic pathways associated with the coupling between aerobic $CH_4$ oxidation and denitrification in paddy soils remain unknown.

Aerobic methanotrophs oxidize $CH_4$ to methanol via $CH_4$ monooxygenase, which is subsequently oxidized by a series of enzymes to formaldehyde, formate, and carbon dioxide ($CO_2$)[19]. As important sources of energy being electron donors, these intermediates may be utilized in combination with diverse terminal electron acceptors including nitrate ($NO_3^-$) and $NO_2^-$[20], which have the potential to stimulate microbial denitrification. For instance, recent studies based on biofilm reactors showed that methanol released during $CH_4$ oxidation promoted the enrichment of methanol-utilizing denitrifiers[16,21]. Meanwhile, aerobic methanotrophs require $O_2$ for growth and metabolic activities, which may form favorable microenvironments following $O_2$ consumption for denitrifiers[22]. Therefore, $CH_4$ oxidation may promote denitrification by the cooperation between aerobic methanotrophs and denitrifiers in hypoxic paddy fields[23]. Furthermore, some aerobic methanotrophs such as *Methylomonas denitrificans FJG1* were capable of partial denitrification with transcription of denitrification genes (e.g., *narG* and *norB*)[24]. Several studies have revealed significant trade-offs between $CH_4$ and nitrous oxide ($N_2O$) emissions in paddy fields[25,26], suggesting possible intertwined associations between $CH_4$ oxidation and denitrification. However, we lack empirical evidence on microbial and metabolic mechanisms involved in the coupling between aerobic $CH_4$ oxidation and denitrification in hypoxic paddy soils. Exploring such knowledge gap can provide valuable insights into N fertilizer management and emission dynamics of the two most important greenhouse gases (GHGs, i.e., $N_2O$ and $CH_4$).

Here, we hypothesized that microbial aerobic $CH_4$ oxidation may promote soil denitrification in rice paddies. Moreover, we propose that this promotion is mediated by the mutualism of diverse microbial species involved in aerobic $CH_4$ oxidation and denitrification. To test these hypotheses, we first conducted a field survey across the major rice-producing regions of China (across a ~3300 km transect) to examine potential correlations between the $CH_4$ oxidation and denitrification activities in paddy soils. We further conducted microcosm experiments to obtain causal insights into the coupling between $CH_4$ oxidation and denitrification. Specifically, we investigated changes in the expression of denitrification genes involved in the complete denitrification pathway including *narG* encoding $NO_3^-$ reductase, *nirK*, *nirS* (encoding $NO_2^-$ reductase), *norB* encoding nitric oxide (NO) reductase, *nosZI* and *nosZII* (encoding $N_2O$ reductase) under the addition of $CH_4$ and aerobic methanotrophs in three distinct paddy soils representative of different soil types. Moreover, we employed $^{13}CH_4$-DNA-stable isotope probing (DNA-SIP) combined with metagenome-assembled genomes (MAGs) and $^{13}C$-metabolomics

analyses to identify key microbial species and metabolic pathways driving the coupling between soil aerobic $CH_4$ oxidation and denitrification.

## Results

### Relation between microbial $CH_4$ oxidation and denitrification in field survey

Our study identified wide variations in denitrification and $CH_4$ oxidation activities as well as in abundances of genes involved in these processes across the main rice-producing areas of China. Specifically, denitrification rates and $CH_4$-oxidizing activities ranged from 0.78 to 164.48 nmol $^{15}N$ $g^{-1}$ soil $h^{-1}$ and 0.31 to 38.81 µg $CH_4$ $g^{-1}$ soil $h^{-1}$, respectively (Fig. 1A). Both denitrification rate and $CH_4$-oxidizing activity were generally lower in the temperate region compared with subtropical and tropical regions of China (Supplementary Fig. 1). Similarly, both abundances of *nirS* and $CH_4$-oxidizing (*pmoA*) genes increased with increasing mean annual temperature ($p < 0.05$, Supplementary Figs. 2 and 3). Importantly, our results showed a significant and positive relationship between denitrification rate and $CH_4$-oxidizing activity ($p < 0.01$). Moreover, the abundance of the *pmoA* gene was positively correlated with the abundances of *nirK* and *nirS* ($p < 0.01$, Fig. 1B). Results of structural equation modeling (SEM) further revealed that $CH_4$-oxidizing activity and *pmoA* abundance had direct and positive associations with denitrification rate, even considering multiple environmental variables ($p < 0.05$, Fig. 1C and Supplementary Fig. 4). Consistently, our correlation network analyses indicated multiple linkages between potential microbial taxa involved in aerobic $CH_4$ oxidation and denitrification (Supplementary Fig. 5). A total of 71 denitrifying and 7 methanotrophic families co-occurred significantly across the major rice-producing areas of China. For example, members of the family *Methylocystaceae* (type II methanotrophs) showed the strongest co-occurrence patterns with denitrifiers, and over 60% of the denitrifiers belonged to *Bradyrhizobiaceae*, *Comamonadaceae*, and *Rhizobiaceae* that had anaerobic or facultative lifestyles (Supplementary Fig. 5).

### Experimental coupling between aerobic $CH_4$ oxidation and denitrification

To verify the influence of aerobic $CH_4$ oxidation on denitrification, we determined changes in $N_2O$ emissions, $NO_3^-$ consumption and the expression of denitrification genes under $CH_4$ addition in the selected soils representative of different soil types (Fig. 2, Supplementary Figs. 6 and 7 and Supplementary Table 1). The addition of $CH_4$ consistently enhanced *pmoA* gene expression, although the community composition of methanotrophs (based on *pmoA* gene transcript) was different in the three types of soils (Supplementary Figs. 8 and 9). Consistently, microcosm experiments showed an overall enhancement in $N_2O$ emissions and $NO_3^-$-N consumption by $CH_4$ addition compared with the control in the selected soils. Specifically, soil $N_2O$ emission was significantly promoted by the addition of 1% $CH_4$ in the black soil and 0.01% $CH_4$ in the red soil ($p < 0.01$). In the yellow soil, $CH_4$ addition consistently promoted $N_2O$ emission by the addition of 0.01% and 1% $CH_4$ ($p < 0.01$, Fig. 2A). The addition of 0.01% and 1% $CH_4$ resulted in an enhanced consumption of $NO_3^-$-N in the black and yellow soils ($p < 0.01$), whereas in the red soil, the promotion was only observed under 0.01% $CH_4$ addition ($p < 0.05$, Supplementary Fig. 7A). Meanwhile, added $CH_4$ generally increased the transcription of *nirK* and *nirS* genes in the three representative soils, with the exception of red soil with 0.1% and 1% $CH_4$ addition (Fig. 2B). Moreover, the addition of $CH_4$ promoted the expression of *nosZI* and *nosZII* in the black and yellow soils, respectively ($p < 0.01$), while 0.01% $CH_4$ addition enhanced the expression of *norB* and *nosZI*, and 1% $CH_4$ addition increased the expression of *nosZI* gene in the red soil ($p < 0.05$, Supplementary Fig. 7B).

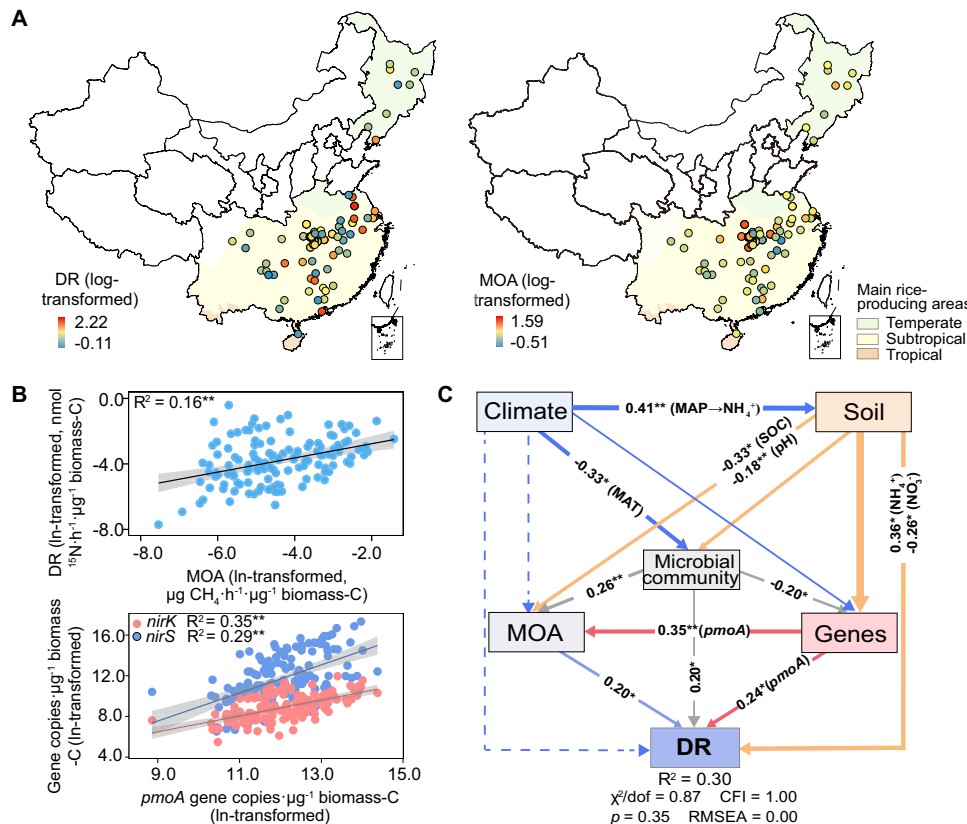

**Fig. 1 | Denitrification rate (DR) and methane-oxidizing activity (MOA) in rice paddy fields across various climatic regions of China. A** The distribution patterns of DR and MOA; **B** Correlations between MOA and DR, and between gene abundances associated with methane oxidation (*pmoA*) and denitrification (*nirK* and *nirS*); **C** Structural equation modeling describing the direct and indirect effects of climate, soil properties, MOA and microbial attributes on DR. The solid lines were fitted by linear regressions in (**B**). The error bands surrounding the regression lines represent the 95% confidence interval of the correlation ($n = 139$). Solid and dashed arrows indicate significant and insignificant relationships in (**C**), respectively. MAP mean annual precipitation, MAT mean annual temperature, SOC soil organic carbon; Microbial community, axes of NMDS ordinations. * Indicates statistically significant levels of $p < 0.05$ and ** indicates $p < 0.01$. We only included direct or indirect associations that could potentially influence DR for graphical simplicity. Additional associations of climate and soil properties with microbial attributes are available in Supplementary Table 6. Exact $p$ values and Source data are provided as a Source Data file.

The addition of methanotrophs also led to an overall increase in $N_2O$ emissions and $NO_3^-$-N consumption compared to the control (Fig. 3A and Supplementary Figs. 6 and 10A). The addition of high amounts of methanotrophs (i.e., $10^{10}$ cell·g$^{-1}$ soil) significantly promoted (4.64 - 13.68 times, $p < 0.05$) the release of $N_2O$ and the consumption of $NO_3^-$-N in the yellow soil, while low amounts of methanotrophs ($2 × 10^9$ cell g$^{-1}$ soil) significantly stimulated these processes in the black soil ($p < 0.01$). However, enhanced $N_2O$ emission was only observed with low amounts of methanotrophs added in the red soil. Similarly, the addition of methanotrophs enhanced the transcript of genes involved in partial or complete denitrification (Fig. 3B and Supplementary Fig. 10B). Specifically, the expression of *nirK*, *nirS* genes increased with increasing amounts of methanotrophs in the black and yellow soils, whereas only the addition of low amounts promoted the expression of *nirK*, *nirS* genes in the red soil ($p < 0.05$). Furthermore, the addition of methanotrophs generally increased the expression of *narG*, *nosZI* and *nosZII* genes in the red soil, whereas it only promoted the expression of *nosZI* gene in the black soil ($p < 0.01$). In the yellow soil, however, the addition of low amounts of methanotrophs enhanced the expression of *nosZI* genes ($p < 0.01$), while the addition of high amounts of methanotrophs increased the expression of *norB* and *nosZII* genes ($p < 0.01$). Linear discriminant analysis effect size (LEfSe) revealed that 32, 27 and 13 operational taxonomic units (OTUs) of *nirS*-denitrifiers varied significantly following the addition of methanotrophs in the black, red and yellow soils, respectively ($p < 0.05$, LDA score >2, Fig. 3C). Importantly, the addition of

methanotrophs consistently enhanced the proportion of *nirK*-denitrifiers belonging to the genus *Ochrobactrum* in the three soils ($p < 0.05$, LDA score >2).

## Microbial guilds associated with the coupling between aerobic CH₄ oxidation and denitrification

We further conducted $^{13}CH_4$-DNA-SIP experiments to identify key taxa associated with the coupling between soil aerobic CH₄ oxidation and denitrification using $^{13}CH_4$ as C sources in the representative soils (Fig. 4 and Supplementary Figs. 6 and 11). The *pmoA* gene initially peaked in the light fractions, and gradually shifted to the heavy fractions on day 15 in all three soils (Supplementary Fig. 11). In contrast, there was a lag in the enrichment of *nirK* and *nirS* genes in the heavy fractions, with significant enrichments on day 40 (Fig. 4A). Results of amplicon sequencing showed that 36, 32 and 28 *pmoA* OTUs were $^{13}C$-enriched in the three soils, respectively (Fig. 4B). For *nirK*-denitrifiers, 35, 110 and 108 OTUs were enriched in the $^{13}C$-labeled heavy fractions, whereas 136, 121, and 259 OTUs of *nirS*-denitrifiers were enriched, respectively.

We found that over 70% of $^{13}C$ labeled *pmoA* OTUs were belonged to the family *Methylocystaceae* (type II methanotrophs) (Fig. 4B and Supplementary Fig. 12). Furthermore, type I methanotrophs (i.e., *Methylomonas*, *Methylomicrobium* and *Methylobacter*) were $^{13}C$-enriched in the black and yellow soils, but no similar enrichments were found in the red soil. Key denitrifiers, including *Rubrivivax* (*nirS* carriers), *Bradyrhizobium* and *Rhizobium* (*nirK* carriers) were

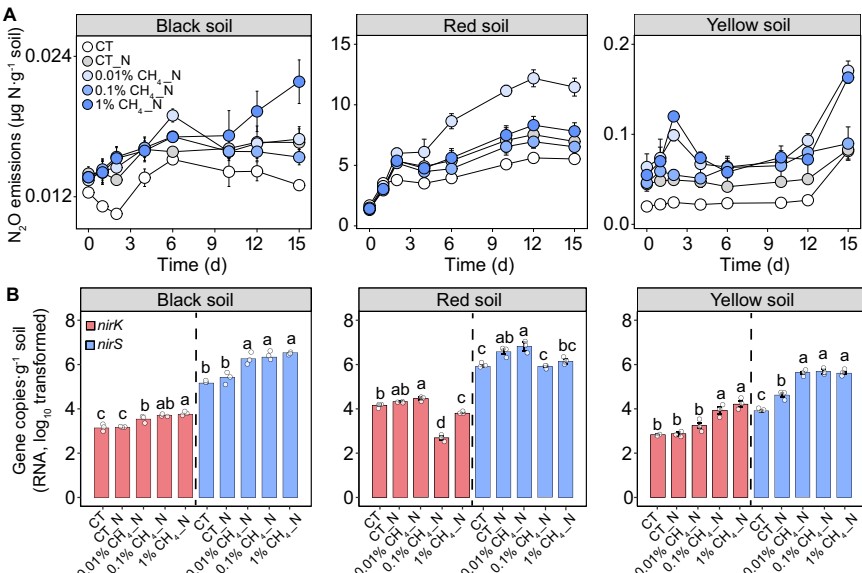

**Fig. 2 | Effects of methane (CH₄) addition on microbial denitrification of three typical paddy soils. A** Variations in nitrous oxide (N₂O) emissions; and **B** Changes in denitrification gene (*nirK* and *nirS*) transcript using RNA reverse transcription. The error bar in (**A**, **B**) represents the standard error of triplicate samples, and data are presented as mean values ± standard error. Different lowercase letters in (**B**) indicate significant differences between the soils with different CH₄ concentrations ($p < 0.05$; $n = 3$; one-way ANOVA followed by two-sided Tukey post hoc test). Exact $p$ values and Source data are provided as a Source Data file.

consistently enriched with $^{13}$C in the heavy fractions of all three soils. Specifically, 22.2–75.2% of $^{13}$C-labeled *nirS* OTUs were belonged to *Rhodospirillaceae* (e.g., *Magnetospirillum*), *Comamonadaceae* (e.g., *Rubrivivax*, *Ideonella*), and *Bradyrhizobiaceae* (e.g., *Bradyrhizobium*) in the three soils. For *nirK*-denitrifiers, 45.0–91.2% of $^{13}$C-labeled OTUs were affiliated with the families *Bradyrhizobiaceae* (e.g., *Bradyrhizobium*, *unclassified Bradyrhizobiaceae*), *Rhizobiaceae* (e.g., *Rhizobium*, *Sinorhizobium*) and *Brucellaceae* (e.g., *Ochrobactrum*).

## Genes and metabolic pathways associated with the coupling between aerobic CH₄ oxidation and denitrification

To further elucidate the metabolic pathways responsible for the coupling between CH₄ oxidation and denitrification, we reconstructed MAGs from $^{13}$C-SIP metagenomics and conducted $^{13}$C-metabolomics experiments (Supplementary Fig. 6). Read assembly and binning from metagenomics recovered a total of 30 MAGs affiliated with methanotrophs and denitrifiers with medium quality (>50% completeness and <10% contamination; Supplementary Table 2; "Methods"). Of these MAGs, 24 fairly-complete genomes (more than 80% completeness, and less than 10% contamination) were further classified as novel species at a 95% average nucleotide identity (ANI) threshold as referred to Genome Taxonomy Database (GTDB) (Fig. 5A, Supplementary Table 2 and Supplementary Figs. 13 and 14). Among these 24 MAGs, 8 were identified as novel species associated with canonical aerobic methanotrophs belonged to the genera *Methylobacter*, *Methylomagnum* and *Methylocystis*. These MAGs generally possessed the entire aerobic CH₄ oxidation pathway, exhibiting the presence of genes responsible for key enzymes such as particulate CH₄ monooxygenase (*pmo*), methanol dehydrogenase (*mxa*, *xoxF*), formaldehyde activating enzyme (*fae*), methylene tetrahydromethanopterin (H₄MPT) dehydrogenase (*mtd*), and formate dehydrogenase (*fdo*, *fdsD*) (Fig. 5A and Supplementary Figs. 13 and 14). In addition to these genes, the results of $^{13}$C-SIP metagenomics at the community level enriched other genes related to CH₄ oxidation, including *mdo* (encoding methanol oxidoreductase involved in the oxidation of methanol to formaldehyde) and *gfa* (encoding S-(hydroxymethyl) glutathione synthase involved in glutathione-dependent formaldehyde oxidation) genes (Supplementary Fig. 13). Furthermore, 5 of these MAGs affiliated with

methanotrophs contained genes related to the partial denitrification process (i.e., *nar*, *nir*, *nor*). For instance, MAGs affiliated with *Methylobacter* (Bin.42 in the black soil) and *Methylocystis* (Bin.56 in the red soil) carried the *nar* gene. However, there were no homologs of the *nosZ* gene in these MAGs.

A total of 16 fairly complete MAGs were identified as novel species affiliated with denitrifiers, which were belonged to genera *Arenimonas*, *Novimethylophilus* and *Magnetospirillum*. These MAGs generally encoded denitrification genes driving the whole denitrification process from NO₃⁻ to N₂, including *nar*, *nap* (encoding NO₃⁻ reductase), *nor*, *nir* and *nosZ* (Fig. 5A and Supplementary Figs. 13 and 14). In particular, *Acidovorax* (Bin.46 in the black soil), *Novimethylophilus* (Bin.52 in the red soil) and *Magnetospirillum* (Bin.19 in the yellow soil) encompassed genes driving the whole denitrification process, including *nap*, *nar*, *nor*, *nir* and *nosZ* genes. In addition, these denitrifying MAGs also enriched genes responsible for the activation of carbonaceous organics such as methanol, formaldehyde, formate, short-chain fatty acids (acetate, propionate, and butyrate), as well as small molecular organic acids involved in pyruvate (lactate), TCA cycle (e.g., malate, and succinate). For instance, *Acidovorax* (Bin.46 in the black soil) and *Magnetospirillum* (Bin.19 in the yellow soil) harbored genes responsible for the activation of acetate (*acs*), propionate (*pct*), lactate (*ldh*), malate (*fum*), and succinate (*suc*).

We subsequently elucidate the $^{13}$C-metabolites derived from the oxidation of $^{13}$CH₄ to support denitrification, through an examination of both the pattern of $^{13}$C labeling fraction and the concentration of each metabolite (Fig. 5B and Supplementary Figs. 6, 15 and 16). We determined various intermediate carbonaceous organics involved in CH₄ oxidation, including formaldehyde, formate, short-chain fatty acids, and intermediates involved in the serine cycle, gluconeogenesis, pyruvate metabolism and TCA cycle (Supplementary Table 3). Overall, the denitrification activity (e.g., the transcription of *norB* and *nosZII* genes) was generally stimulated by the addition of NO₃⁻ (Supplementary Fig. 7). In the three representative soils, we observed significant variations in the $^{13}$C-labeled fraction of short-chain fatty acids (e.g., acetate, propionate, and butyrate) and small molecular organic acids involved in pyruvate (lactate) and TCA cycle (e.g., malate, and succinate) across different treatments (Fig. 5B). Specifically, the addition of

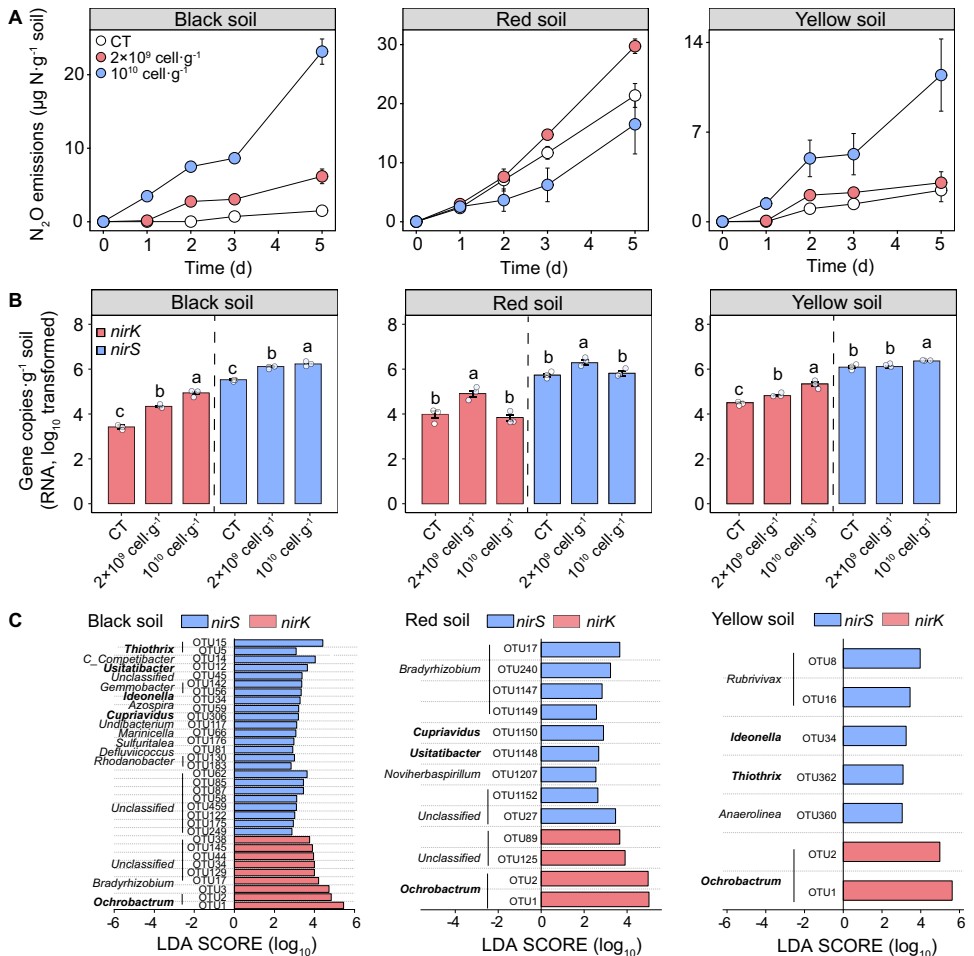

**Fig. 3 | Effects of the addition of aerobic methanotrophs on soil denitrification.** **A** Variations in nitrous oxide (N₂O) emissions; **B** Changes in the transcript of denitrification genes (*nirK* and *nirS*); **C** Biomarkers of key *nirK*-denitrifiers and *nirS*-denitrifiers at the genus level (based on RNA reverse transcription) in methanotrophs addition treatment. The error bar in (**A**, **B**) represents the standard error of triplicate samples, and data are presented as mean values ± standard error.

Different lowercase letters in (**B**) indicate significant differences between soils with different amounts of aerobic methanotrophs added ($p < 0.05$; $n = 3$; one-way ANOVA followed by two-sided Tukey post hoc test). The biomarkers in (**C**) were identified by linear discriminant analysis effect size method ($p < 0.05$, LDA score >2). Exact $p$ values and Source data are provided as a Source Data file.

¹³CH₄ generally increased the ¹³C-labeled fraction of these short-chain fatty acids and organic acids in the black and yellow soils compared with the control treatment. The simultaneous addition of ¹³CH₄ and NO₃⁻ addition resulted in an overall increase in the ¹³C-labeled fraction of acetate, propionate, lactate, malate, and succinate in the black soil, compared with the addition of ¹³CH₄ ($p < 0.05$). Similarly, elevated ¹³C-labeled fractions of propionate, succinate and lactate were observed in the yellow soil ($p < 0.05$). In contrast, the combined addition of ¹³CH₄ and NO₃⁻ led to an overall decline in the concentration of acetate, propionate and butyrate in the black and yellow soils, as compared to the ¹³CH₄ treatment. In the red soil, however, the simultaneous addition of ¹³CH₄ and NO₃⁻ reduced ¹³C fractions but increased concentrations of acetate, propionate, and butyrate, compared with the addition of ¹³CH₄ ($p < 0.05$, Supplementary Fig. 16). Other intermediates involved in CH₄ oxidation, such as formaldehyde, formate, and carbonaceous organics associated with the serine cycle and gluconeogenesis, did not exhibit enrichment in ¹³C or significant variations across different treatments, despite the variations in their concentrations (Supplementary Figs. 6 and 15).

## Discussion

Our study provides valuable insights into the coupling between microbial aerobic CH₄ oxidation and denitrification in paddy fields

harboring diverse aerobic and anaerobic taxa, demonstrating ubiquitous linkages of these two fundamental processes through species and metabolic couplings in soils in various climatic regions. Our results showed significant and positive correlations between soil microbial CH₄ oxidation and denitrification activities across major rice-producing areas of China. The facilitated soil denitrification by the addition of CH₄ or aerobic methanotrophs confirms potential interactions between aerobic CH₄ oxidation and denitrification. Additionally, we identified key species involved in the coupling of aerobic CH₄ oxidation and denitrification using ¹³C-DNA-SIP. Our ¹³C-MAGs and ¹³C-metabolomics measurements further indicate that denitrifiers could utilize intermediate compounds derived from aerobic CH₄ oxidation, such as acetate, propionate, butyrate and lactate, thereby enabling the coupling between aerobic CH₄ oxidation and denitrification. While similar coupling has been found in pure culture systems and bioreactors[27,28], our work provides the integrated assessment of the relative importance of different C sources for denitrification by implementing ¹³CH₄-metabolomics within complex soil systems. Furthermore, our findings shed light on previously unexplored biochemical interactions in paddy fields and hold significant implications for improving the prediction of GHG emissions in these economically and socially important agroecosystems.

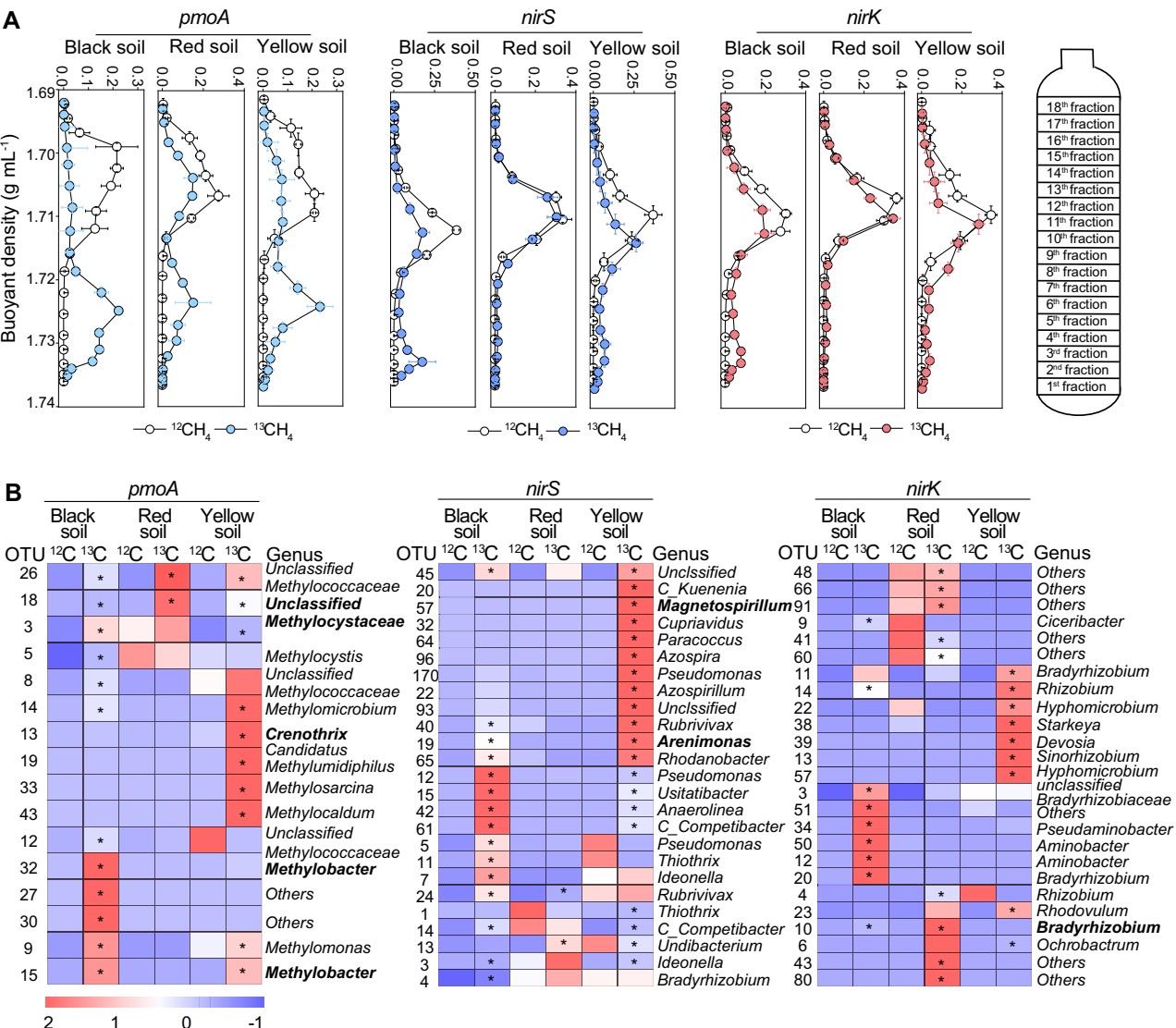

**Fig. 4 | Bacterial taxa associated with genes involved in methane (CH$_4$) oxidation (*pmoA*) and denitrification (*nirS* and *nirK*) in $^{13}$C-CH$_4$ amended microcosms. A** Quantitative distribution of the *pmoA*, *nirS* and *nirK* genes in 18 fractions from each CsCl gradient, covering a buoyant density range from 1.69 to 1.74 g ml$^{-1}$; **B** The relative abundance (*Z*-score standardized) of the top 25 genera within the $^{13}$C-heavy fractions. Gene reads are derived from the entire buoyant density gradient of DNA fractions extracted from the three typical soils incubated with $^{12}$CH$_4$ and $^{13}$CH$_4$ for 40 days. The error bar in (**A**) represents the standard error of the triplicate samples (*n* = 3), and data are presented as mean values ± standard error. * in (**B**) indicates operational taxonomic units (OTUs) that exhibited significant enrichment in the heavy fractions of the $^{13}$C-labeled treatment compared to the heavy fractions in the corresponding $^{12}$C control based on *Z*-score. Source data are provided as a Source Data file.

The positive associations between the potential activity of CH$_4$ oxidizers and denitrifiers (including both activities and functional genes) are in agreement with our hypothesis and indicate a prevalent co-occurrence of the two processes in paddy fields along the transect across major rice-producing areas of China. Our SEM analysis provides further evidence that CH$_4$-oxidizing activity and the abundance of the *pmoA* gene play significant roles in influencing denitrification rate, even after considering multiple environmental variables. Additionally, the proportions of key methanotrophs (e.g., *Methylomonadaceae*, unclassified *Methylococcales* and *Methylocystaceae*) showed positive relationships with denitrification rate and denitrification genes, suggesting a significant promotion of denitrification activity in sites with abundant methanotrophs (Supplementary Fig. 17). These results highlight the importance of CH$_4$ oxidation in predicting soil denitrification (Fig. 1C), a process typically attributed to the regulation of canonical denitrifiers and genes (e.g., *nirK* and *nirS*)[29,30]. These findings offer significant insights into the denitrification process from the

perspective of elemental coupling, indicating that the facilitation of denitrification by CH$_4$ oxidation may be a potentially important pathway, linking C and N cycling.

The coupling of CH$_4$ oxidation with soil denitrification was further validated by the results of experiments supplementing representative paddy soils with CH$_4$ and methanotrophs addition (Figs. 2 and 3). The increased N$_2$O production and the consumption of NO$_3^-$-N indicate a significant promotion of soil denitrification by CH$_4$ oxidation. Similar NO$_3^-$ removal by CH$_4$ addition has been observed in well-designed biofilm bioreactors, where CH$_4$ was introduced as the sole source of C[31]. Although the variations in NO$_3^-$ removal efficiency under different NO$_3^-$ and O$_2$ concentrations in these bioreactors have been extensively studied[22,32], limited knowledge exists regarding the specific step in denitrification through which CH$_4$ oxidation is coupled to the denitrification process. The generally enhanced expression of *nirK* and *nirS* genes upon the addition of CH$_4$ or methanotrophs in the three selected soils suggests that NO$_2^-$ might serve as a substrate for aerobic CH$_4$

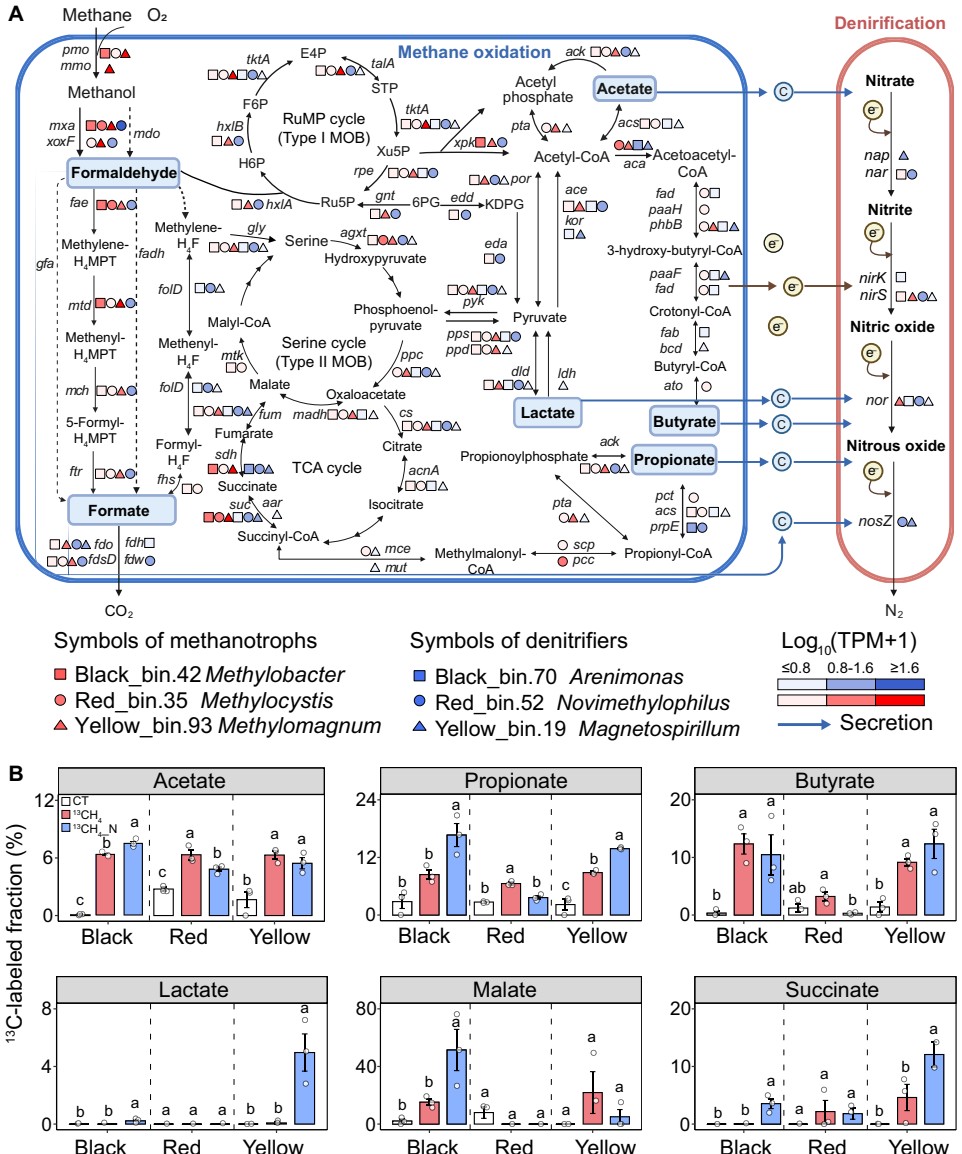

**Fig. 5 | The proposed metabolic pathways of the coupling between aerobic methane (CH$_4$) oxidation and denitrification in the paddy soils. A** The color gradients represent the logarithm of transcripts per million (TPM) of major genes in the corresponding metagenome-assembled genomes (MAGs) classified as methanotrophs and denitrifiers, respectively, in the heavy DNA from $^{13}$CH$_4$ incubation. The definitions of genes are listed in Supplementary Table 7. **B** Changes in the $^{13}$C-labeled fraction of metabolites derived from $^{13}$CH$_4$ oxidation in the three typical soils. The square, circle, and triangle in (**A**) represent MAGs recovered from black soil, red soil, and yellow soil, respectively. The error bar in (**B**) represents the standard error of triplicate samples, and data are presented as mean values ± standard error ($n = 3$; one-way ANOVA followed by two-sided Tukey post hoc test). Different lowercase letters in (**B**) indicate significant differences between treatments ($p < 0.05$). Exact $p$ values and Source data are provided as a Source Data file.

oxidation coupled to denitrification in these soils (Figs. 2 and 3). Moreover, the promoted expression of *narG* and *nosZII* genes with CH$_4$ addition in the black and yellow soils implies that aerobic CH$_4$ oxidation could also be coupled with the denitrification process using NO$_3^-$ and N$_2$O as substrates (Supplementary Fig. 7). In the red soil, however, the enhanced expression of *norB* and *nosZI* genes by addition of CH$_4$ and *narG* and *nosZII* genes by addition of methanotrophs indicates the coupling of aerobic CH$_4$ oxidation with denitrification using NO$_3^-$, NO and N$_2$O as substrates (Supplementary Figs. 7 and 10). These results highlight variations in the specific steps of the coupling between CH$_4$ oxidation and denitrification across different soils, thereby providing a more mechanistic understanding of this interaction of C and N cycles.

Flooded paddy fields constitute structured ecosystems characterized by a partially oxic or hypoxic surface layer, due to the leakage of O$_2$ from the roots of rice plants[12]. The hypoxic environment provides

a favorable environment for the coexistence of various aerobic and anaerobic microbes, including diverse aerobic methanotrophs and denitrifiers[12]. In these hypoxic conditions, the decomposition of organic matter commonly results in the accumulation of CH$_4$, with soil gas and pore water containing up to ~14% and 550 μM of dissolved CH$_4$, respectively[33]. Previous studies suggest that over 70% of endogenous CH$_4$ can be consumed by soil aerobic methanotrophs in hypoxic conditions before escaping to the atmosphere[14,15], serving as a potential link for the coupling between CH$_4$ oxidation and denitrification. Supporting this potential connection, we found a significant co-occurrence between methanotrophs and denitrifiers across the national-scale transect of paddy fields. We posit that this significant co-occurrence could be attributed to the niche similarity between these functional groups, as well as the possibility of energy and microbial species interactions[8,34]. For instance, denitrifiers may benefit from the

$O_2$ depletion associated with $CH_4$ oxidation, creating temporal anoxic microenvironment that favors denitrifiers[21,35]. This assumption could be supported by the particularly strong co-occurrence between aerobic methanotrophs and anaerobic denitrifiers (Supplementary Fig. 5). Additionally, the intermediate carbonaceous organics released during aerobic $CH_4$ oxidation can serve as C sources for the denitrification process[27,28,36]. In support of this hypothesis, we found that the addition of aerobic methanotrophs significantly enriched denitrifiers carrying *nirK* and *nirS* genes (Fig. 3). Many of these enriched denitrifiers such as *Ochrobactrum*, *Thiothrix*, and *Bradyrhizobium* can aerobically or anaerobically utilize small molecules from microbial $CH_4$ oxidation (e.g., acetate, propionate, butyrate and lactate)[37–42]. Moreover, our $^{13}CH_4$-DNA-SIP results revealed significant enrichments of $^{13}C$ in denitrifiers carrying the *nirK* and *nirS* genes, indicating assimilation of $^{13}C$ derived from $^{13}CH_4$ by denitrifiers (Fig. 4). Importantly, the identified methanotrophs (e.g., unclassified *Methylocystaceae*) and denitrifiers (e.g., *Bradyrhizobium*, *Magnetospirillum*, *Rubrivivax*, *Ideonella*) enriched $^{13}C$ also showed the strongest co-occurrence patterns across distinct soil types following the field survey. These results indicate that aerobic methanotrophs can affect denitrification activity by selectively recruiting specific denitrifying groups, and highlight the prevalence of the interactions between these two functional guilds in paddy fields.

Further $^{13}CH_4$-SIP-metagenomic and $^{13}C$-metabolomics results indicate that multiple metabolic pathways could contribute to the coupling between $CH_4$ oxidation and denitrification (Fig. 5). This is evidenced by the co-occurrence of various $CH_4$-oxidizing and denitrification genes in the $^{13}C$-heavy fractions, including *mdo*, *fae*, *nap* and *nosZ* genes. Moreover, our results of MAGs indicate that essential denitrifiers, such as Species Genome Bins (SGBs) belonged to genera *Acidovorax*, *Magnetospirillum* and *Novimethylophilus*, significantly enriched C-utilization genes including *acs*, *pct*, *ldh*, *fum*, *suc*, *sdh*, *fae*, and *fdo* (Fig. 5A and Supplementary Figs. 13 and 14). The enrichments of C-utilization genes indicate that these denitrifiers possess the metabolic potential to utilize intermediates released during $CH_4$ oxidation, including short-chain fatty acids (acetate, propionate, and butyrate), organic acids involved in pyruvate (lactate), TCA cycle (e.g., malate, and succinate), formaldehyde, and formate. However, none of the identified denitrifiers carried *pmo* and *mmo* genes (encoded soluble $CH_4$ monooxygenase), indicating that these $^{13}C$-enriched denitrifiers likely assimilate the $^{13}C$ by utilizing intermediates released from methanotrophs, rather than directly utilizing $CH_4$. In support of this hypothesis, we observed an overall higher $^{13}C$-labeled fraction of short-chain fatty acid (e.g., acetate, propionate, and butyrate) as well as organic acids involved in pyruvate metabolism and TCA cycle (e.g., lactate) under the simultaneous addition of $^{13}CH_4$ and $NO_3^-$, as compared to the solely adding $^{13}CH_4$ in the black and yellow soils. It is assumed that denitrifiers would preferentially utilize the lighter $^{12}C$ initially, thereby leading to the enrichment of heavier $^{13}C$ in the residual substrate[43,44]. Therefore, the overall higher $^{13}C$-labeled fraction of acetate, propionate, butyrate, and lactate indicates an increased consumption of these intermediates in response to the stimulation of denitrification activity by $NO_3^-$-N addition. This notion could be further supported by the overall lower concentration of these intermediates under the simultaneous $^{13}CH_4$ and $NO_3^-$ addition in these soils. While the potential of certain $CH_4$ oxidation intermediates to facilitate denitrification has been recognized in bioreactor studies[45,46], our research expands this knowledge by providing direct $^{13}C$-metabolomic evidence, with essential implications on the relative importance of different carbon sources in supporting denitrification within complex soil systems in paddy fields.

However, no such enrichment of $^{13}C$-labeled metabolites was observed in the red soil, where methanotrophs were predominantly composed of (over 90%) type II methanotrophs. Previous studies have reported that type II methanotrophs, including *Methylocystis* and *Methylosinus*, were less tolerant to high N concentration[47,48].

Therefore, the effects of $NO_3^-$ on methanotrophs may exert confounding effects on the coupling between $CH_4$ oxidation and denitrification. Nevertheless, the higher consumption of short-chain fatty acids (e.g., acetate, propionate, and butyrate) and organic acids involved in pyruvate metabolism and TCA cycle (e.g., lactate) in response to increased denitrification indicates that these intermediates could serve as C sources for denitrifiers, thereby facilitating the coupling between $CH_4$ oxidation and denitrification. It is also plausible that certain methanotrophs possess homologs of denitrification genes, such as *nirS*, *nirK*, and *norB*[24], which may also contribute to the coupling between $CH_4$ oxidation and denitrification. However, SGBs of methanotrophs did not identify homologs of *nosZ* gene, consistent with the findings from previous studies[17,49]. Therefore, the elevated expression of *nosZ* gene in response to addition of $CH_4$ and methanotrophs underscores the influence of $CH_4$ oxidation on denitrifiers, rather than the expression of denitrification genes within methanotrophs themselves, playing pivotal roles in the coupling between $CH_4$ oxidation and denitrification. These results provide direct evidence for the coupled metabolisms of aerobic $CH_4$ oxidation and denitrification, highlighting the importance of the species interactions for the denitrification process in paddy soils. Nevertheless, we acknowledge that the specific coupling pathways of $CH_4$ oxidation and denitrification may vary among different strains and within diverse microbial communities. For instance, the results from $^{13}CH_4$-SIP indicate that *Rhizobia*, as carriers of denitrification gene *nirK*, have the potential to utilize C from $CH_4$, but with specific metabolic pathways remaining unclear[50,51]. Therefore, additional studies, such as isolating these specific taxa in pure culture and utilizing third-generation metagenomic approaches to obtain higher quality MAGs, are warranted to reinforce and consolidate the findings presented in this study.

It is also noteworthy to mention that the magnitude of increase in denitrification activity promoted by aerobic $CH_4$ oxidation varied across representative soils of different climate regions. These different responses could be partly explained by differences in soil organic C (SOC). Substrate availability has long been proposed to be responsible for biotic interactions between microbial species[52]. In support of this, we found that variations in SOC were important predictors for the relationship between denitrification and $CH_4$ oxidation activity (Supplementary Fig. 18). In the C-rich soils from temperate regions, the addition of low-concentration $CH_4$ had little effect on denitrification activity, because small organic molecules produced during $CH_4$ oxidization could be negligible compared with the indigenous soil C pool. In contrast, denitrification activities were significantly promoted with the addition of low-concentration of $CH_4$ in the low-C soils from subtropical and tropical regions. Furthermore, differences in the relative abundance of various species in the indigenous community may also contribute to the different responses of denitrification activity to $CH_4$ addition. For instance, the red soil harbored high proportion of type II methanotrophs (~95%), which have life strategies with slow growth rates[53] and thus may account for the weak response of denitrification to high $CH_4$ addition (Supplementary Fig. 8). High amounts of $CH_4$ may lead to rapid depletion of $O_2$ and nutrients[36], confounding $CH_4$ effects on denitrification in paddy soils. Taken together, these results suggest a high sensitivity of denitrification activity with $CH_4$ accumulation in the low-C soils from subtropical and tropical regions. Intriguingly, our field surveys indicate that these regions are hotspots for both $CH_4$ oxidation and denitrification. These observations reveal that the accumulation of $CH_4$ even in small amounts has the potential to significantly facilitate denitrification in paddy soils from these regions.

In summary, our work identified key microbial taxa and metabolic pathways driving aerobic $CH_4$ oxidation coupled with denitrification in paddy soils. Importantly, we find that over 70 phylotypes involved in denitrification are associated with the assimilation of $^{13}C$ derived from $CH_4$. We identified crucial genes and metabolic processes associated

with the coupling between $CH_4$ oxidation and denitrification. We highlight the importance of intermediates produced during aerobic $CH_4$ oxidation, such as acetate, propionate, butyrate and lactate, in facilitating the coupling of $CH_4$ oxidation with denitrification. Additionally, we show significant co-occurrence of these identified aerobic methanotrophs and denitrifiers across major rice-producing areas of China, highlighting the importance of species interactions between the two functional guilds in regulating soil denitrification in paddy agroecosystems. This knowledge is critical to improving predictions of the flux of GHG emissions in agroecosystems under global change scenarios.

## Methods

We combined a field survey with a series of microcosm experiments using soils representative of different soil types to investigate the role of soil aerobic $CH_4$ oxidation in driving denitrification (Supplementary Fig. 6). To provide an overview, we first conducted a field study to explore potential connections between $CH_4$ oxidation and denitrification activities in paddy soils across the major rice-producing regions of China. Subsequently, we delved into the co-occurrence patterns of microbial taxa involved in $CH_4$ oxidation and denitrification in these paddy soils across the transect, to identify potential microbial taxa involved in these two processes. To further substantiate the relationship between microbial aerobic $CH_4$ oxidation and denitrification in the field survey, we conducted microcosm experiments using three soils that were representative of different soil types (i.e., black, red, and yellow soils). In these experiments, both $CH_4$ and methanotrophs were added to assess the influence of $CH_4$ oxidation on the activity and gene transcription of denitrification. Furthermore, we carried out $^{13}CH_4$-DNA-SIP experiments, encompassing the following components: (1) identifying crucial microbial taxa involved in $CH_4$ oxidation and denitrification through amplicon sequencing; (2) investigating key functional genes related to $CH_4$ oxidation and denitrification using metagenomics sequencing; and (3) examining the potential for metabolic coupling between $CH_4$ oxidation and denitrification within methanotrophs and denitrifiers by reconstructing MAGs from $^{13}C$-SIP metagenomics data. Finally, we conducted $^{13}C$-metabolomics experiments to identify the metabolic intermediates and validate the metabolic pathways associated with $CH_4$ oxidation and denitrification.

### Field survey and sampling

Soil samples were predominantly collected in 2018 (from November 2017 to March 2019) from the main rice-producing areas of China, with latitude ranging from 19.99 °N to 47.24 °N and longitude ranging from 102.74 °E to 130.59 °E (across a >3300 km transect), accounting for over 90% of total rice production in the temperate, subtropical and tropical regions of the country (Supplementary Fig. 1A). These areas span large climatic gradients, with the mean annual precipitation and mean annual temperature ranging from 493 to 1861 mm and 2.9 °C to 25 °C, respectively. Our sampling strategy was designed to ensure extensive coverage of the environmental gradients. At each microsite, five 1 m × 1 m sub-plots were located at each corner and center of a 50 m × 50 m area. Five soil cores (top 15 cm) were collected and then mixed to a composite soil sample to consider the heterogeneity. Samplings were conducted under waterlogged conditions. Detailed information on sampling strategy was shown in Supplementary Fig. 6. The collected soil samples were sealed in a polyethylene bag and sent back to laboratory on ice in cooling boxes. We separated each soil sample into three subsamples, one subsample was kept at 4 °C for analyses of denitrification rate and $CH_4$-oxidizing activity, one subsample was freeze-dried and used for microbial DNA extraction and bacterial community analyses, while the remaining samples were air-dried for soil chemical analyses.

### Soil microbial activity and chemical analyses

Denitrification rate in paddy soils was measured using the $^{15}N$ isotope-pairing technique[54,55]. Briefly, 5.0 g fresh soil was transferred to a 50 ml sterilized anaerobic bottle. The bottles were purged with high-purity He (99.99%) for 10 min to replace the headspace air. The samples were first pre-incubated at 20 °C for 24 h to remove residual $NO_3^-$. Subsequently, the bottles were spiked with $K^{15}NO_3$ solution (99.8% atom, Macklin Co. China) to reach a final concentration of 50 μg N g$^{-1}$ soil, and then re-purged with high-purity He. The samples were incubated in the dark for 8 h at 20 °C. After the incubation, the headspace gas samples were transferred to 12 ml vials. The concentrations of $^{28}N_2$, $^{29}N_2$, and $^{30}N_2$ in the headspace were quantified with an isotope ratio mass spectrometry (MAT 253 plus, Thermo, USA). The denitrification rate was calculated as follows:

$$\text{Denitrification rate}\left(\text{nmol g}^{-1}\, soil\right) = \frac{T_{30} \times 2 \times (1 - F_n)}{F_n} - 2 \times T_{30} \quad (1)$$

where $T_{30}$ (nmol $^{30}N_2$ l$^{-1}$ h$^{-1}$) represents the production rates of $^{30}N_2$ during the 8 h. $F_n$ (99%) is the $^{15}N$ fraction in added $^{15}NO_3^-$ after preincubation.

For the determination of $CH_4$-oxidizing activity, 5.0 g fresh soil was weighed to a 120 ml serum bottle and then 1% (v/v) high-purity $CH_4$ (about 1.2 ml) was injected to the sealed bottles[53]. Headspace gases were sampled at 0 and 2 h because $CH_4$ was consumed linearly during 2 h according to our pre-incubation experiments in representative soil samples. Similarly, previous studies indicated that $CH_4$ oxidation rate remained relatively constant within 8 h of incubation[56,57]. The $CH_4$ concentration in the headspace gas was determined using an Agilent GC7890A gas chromatograph equipped with a flame ionization detector and an electron capture detector (Agilent Technologies, Wilmington, DE, USA), and the flame ionization detector was used for $CH_4$. The $CH_4$-oxidizing activity was calculated as μg $CH_4$ g$^{-1}$ soil h$^{-1}$.

Soil pH was measured by a Delta pH-meter (Mettler-Toledo Instruments Co., Columbus, OH, USA) in a soil slurry with a water-to-soil ratio of 2.5:1. SOC was determined by potassium dichromate oxidation titration method[58]. Ammonium and $NO_3^-$ were extracted with 2 M Potassium chloride and measured by a FIAstar 5000 Analyzer (Foss Tecator, Denmark). Microbial biomass carbon was measured using the fumigation-extraction method[59].

### Microbial community analyses

Microbial DNA was extracted from 0.25 g freeze-dried soil using the MoBio PowerSoil DNA Isolation Kit (MoBio Laboratories, Carlsbad, CA, USA) according to the instructions. The concentration and quality of extracted DNA were checked by a nanodrop-2000c UV VIS spectro-photometer (nanodrop technologies, Wilmington, DE, USA). Key functional genes responsible for $CH_4$ oxidation (pmoA gene) and denitrification (nirK and nirS genes) were quantified in a 20 μl PCR mixture using Real-Time quantitative PCR System (qPCR, LightCycle 480 II, America). The primer pairs and thermal cycling parameters were provided in Supplementary Table 4. Standard curves were generated using serial dilutions (10-fold) of the plasmid of genes. Standard curves had an amplification efficiency ranging from 81.6% to 96.9% and the linear correlation coefficient ($R^2$) of genes ranged from 0.990 to 1.000.

Bacterial community composition was characterized by Illumina MiSeq sequencing of the V3-V4 region of 16S rRNA gene amplified using the primer pairs of 338F (5′-ACTCCTACGGGAGGCAGCA-3′) and 806R (5′-GGACTACHVGGGTWTCTAAT-3′)[60]. The barcoded PCR products were purified using the Wizard SV Gel and PCR Clean-Up System (Promega, San Luis Obispo, CA, USA). The purified amplicons were equimolarly mixed, and 2 × 300 bp paired-end sequencing was carried

out on an Illumina Miseq sequencer (Illumina Inc., San Diego, CA, USA). Raw and paired-end sequences were quality filtered and assembled using the Fast Length Adjustment of Short Reads (FLASH) software (v1.2.11)[61]. The OTU was partitioned at 97% sequence similarity using the UPARSE (v7.1)[62]. Representative sequences from individual OTUs generated in UPARSE were processed using the Quantitative Insights into Microbial Ecology (QIIME) pipeline (v2020.2)[63]. Shifts in the bacterial community composition were determined by using two axes of NMDS analysis of Bray-Curtis dissimilarities. Taxonomy assignments of bacterial phylotypes were performed in reference to the SILVA (https://www.arb-silva.de/documentation/release-128/, v 128)[64].

### Microcosm experiments

**$CH_4$ addition experiments.** To verify the relation between microbial aerobic $CH_4$ oxidation and denitrification in the field survey, we conducted $CH_4$ addition experiment in three soils representative different soil types (i.e., black, red, and yellow soils). The geographical information and physicochemical characteristics of three typical soils were shown in Supplementary Table 1. Specifically, we investigated the response of denitrification functions involved in $N_2O$ production, $NO_3^-$ consumption and gene expression under different concentration of $CH_4$ addition. The determined denitrification genes encompassed those associated with complete denitrification pathways, including *narG*, *nirK*, *nirS*, *norB*, *nosZI* and *nosZII* genes. Prior to execution of the microcosm experiments, the water saturated soil was pre-incubated at 25 °C in the dark for a week under ambient air conditions. Then, 10.0 g of soil was mixed with 15 ml water, 1 mg potassium nitrate ($KNO_3$-N) and 0.01 mg 3,4-dimethylpyrazole phosphate (DMPP), and placed into a 120 ml serum bottle, capped with butyl rubber stoppers (three replicates). $KNO_3$ was provided as the substrate for denitrification and DMPP was used to inhibit nitrification and ensure that $N_2O$ was predominantly released by denitrification[65]. On average, $N_2O$ production was reduced by 81.1% (ranging from 70.7–91.5%) with DMPP addition (Supplementary Fig. 19), which is comparable to several previous studies[66-68]. The headspace gas was flushed with pure air (80% $N_2$, 20% $O_2$) and different amounts of $CH_4$ were added to obtain initial $CH_4$ concentrations of 0, 0.01%, 0.1% and 1% (v/v). All samples were incubated at 25 °C in the dark for 15 days. Headspace gas was collected periodically to analyze the concentrations of $N_2O$, $CH_4$ and $CO_2$ by gas chromatograph on days 0, 1, 2, 4, 6, 10, 12, 15, respectively. At the end of the incubation, soil samples were quickly frozen with liquid nitrogen and stored at −80 °C prior to RNA extraction. Total RNA was extracted from 2 g fresh soil using the RNeasy PowerSoil Total RNA Kit. The concentration and quality of extracted RNA were checked by a nanodrop-2000c UV VIS spectrophotometer and agarose gel electrophoresis. Notably, clear bands corresponding to 28S, 18S, and 5S were observed in all samples, confirming the high quality of the total RNA. The total RNA was converted to complementary DNA (cDNA) using a PrimeScript™ RT reagent Kit with gDNA Eraser (TaKaRa), and was stored at −20 °C for quantifying denitrification and *pmoA* genes and amplicon sequencing of *pmoA* gene. To prevent contamination with DNA, RNA fraction was applied to DNase treatment, and 2 µl of RNA was extracted as contamination control after DNase treatment and before cDNA synthesis. The extracted RNA was then subjected to PCR amplification to check for the presence of any DNA contamination. Importantly, no PCR products were observed in any of the RNA samples after DNase treatment, providing solid evidence for the absence of DNA contamination.

**Addition of aerobic methanotrophs.** Typical aerobic methanotrophs (*Methylosinus trichosporium*) were added to investigate the effect of methanotrophs on denitrification. *M.trichosporium* was purchased from NCIMB (National Collection of Industrial, Food and Marine, England). Specifically, a typical species *M. trichosporium* of the family *Methylocystaceae* was selected because it showed the strongest co-

occurrence patterns with denitrifiers in our field survey. *M. trichosporium* was grown in $NO_3^-$ minimal salts medium at 30 °C[69]. To ensure a high percentage of active cells, we harvested cells in the late exponential phase, washed three times, and then resuspended in 5 mM phosphate-buffered saline solution at pH 7.3[70]. Three levels of cell numbers (0, $2 \times 10^9$ and $10^{10}$ cell $g^{-1}$ soil) were added to 10.0 g of soil and medium (15 ml water, 1 mg $KNO_3$-N and 0.01 mg DMPP) for 5 days following previous studies[71,72]. The headspace gas was collected and measured as stated above on days 0, 1, 2, 3, 5. On day 5, soil samples were collected for RNA extraction and reverse transcription. The cDNA was used to quantify *narG*, *norB*, *nirK*, *nirS*, *nosZI*, *nosZII*, *pmoA* genes, and to sequence *nirK*, *nirS* amplicons.

**$^{13}CH_4$-DNA-stable isotope probing experiments.** The DNA-SIP was conducted to identify the microbial taxa responsible for the coupling between aerobic $CH_4$ oxidation and denitrification using $^{13}C$-labeled or unlabeled $CH_4$ as C sources. The incubations comprised of 10.0 g of soil, 15 ml water, 1 mg $KNO_3$-N and 0.01 mg DMPP. Soil cultures were incubated with 10% $CH_4$ (v/v, labeled with 99.9% $^{13}C$) as previously described[73-75]. This concentration was chosen to represent the extensive accumulation of $CH_4$ associated with organic matter decomposition in anoxic conditions[33], and to ensure the validity of results for SIP experiments. Previous studies have indicated that wetland environments, such as rice paddies, often exhibit substantial $CH_4$ production due to hypoxic conditions and high level of organic matter accumulations[33]. Within these high SOC and hypoxic systems, previous studies indicated that monthly in situ porewater dissolved $CH_4$ concentrations in mud and water-covered soils exceeded 0.15 mM[33]. This $CH_4$ concentration in the pore water could be obtained by an initial $CH_4$ concentration of 10% (v/v) in the air, calculated by Henry's Law and the van't Hoff equation to account for temperature-dependent solubility of $CH_4$[76]. Furthermore, $CH_4$ was replenished when 90% of the $CH_4$ was consumed[19,77]. All microcosms were incubated at 25 °C in the dark (three replicates), and were destructively sampled on days 0, 15 and 40[78]. Microbial DNA was extracted from 0.25 g freeze-dried soil using the MoBio PowerSoil DNA Isolation Kit (MoBio Laboratories, Carlsbad, CA, USA) following the instructions. The DNA extraction procedure with this kit included mechanical and chemical lysis of cells and subsequent column-based DNA purification[79]. Genomic DNAs were separated into heavy (i.e., $^{13}C$-DNA) fractions and light (i.e., $^{12}C$-DNA) fractions by CsCl gradient ultracentrifugation. Briefly, ~5000 ng genomic DNA was mixed with CsCl and gradient buffer in 5.1 ml OptiSeal polyallomer tubes (Beckman Coulter, Palo Alto, USA) to achieve an initial buoyant density (BD) of 1.713 g $ml^{-1}$. The mixture was centrifuged at 176,770 × g for 45 h at 20 °C in an Optima XPN-100 Ultracentrifuge (Beckman Coulter) equipped with a V65.2 vertical rotor (Beckman Coulter, USA). The obtained DNA gradients were fractionated into 18 equal volumes (~250 µl). The BD of each fraction was measured by determining the refractive index with a digital refractometer (Palette, ATAGO, Japan). The fractionated DNA was recovered by polyethylene Glycol 6000 (PEG6000) and then recovered with 70% absolute ethanol and finally eluted with 40 µl of TE buffer (pH 8.0).

**Quantification and sequencing of genes involved in $CH_4$-oxidation and denitrification.** The relative proportion of putative $CH_4$ oxidizers and denitrifiers in each fraction to the whole gradient was determined by qPCR of the *pmoA*, *nirS* and *nirK* genes from each treatment. The pooled heavy fractions recovered DNA from the $^{12}CH_4$- and $^{13}CH_4$-incubated samples were used for *pmoA*, *nirS* and *nirK* gene amplicon sequencing using the Illumina MiSeq and pair-end 300 bp mode at Majorbio in Shanghai, China. Raw sequences were analyzed with QIIME2, and the curated sequences were clustered into OTUs with an 85% similarity using vsearch[80]. The selection of a similarity threshold is a critical decision that necessitates a careful balance to prevent both

the excessive aggregation of functionally divergent sequences and the insufficient clustering of sequences that are functionally similar. Previous research has established that a similarity threshold within the range of 80–90% effectively balances biological diversity and conservatism for genes involved in $CH_4$ oxidation and denitrification processes[81,82]. To identify the most suitable similarity threshold in our study, OTU clustering at various thresholds (spanning from 60% to 97%) was systematically conducted, and an inflection point was observed in the OTU count at the 85% similarity threshold (Supplementary Fig. 20). This threshold was selected as it best met our analytical needs, taking into account the specific features of the soil ecosystem and the functional genes under investigation. Therefore, we used the 85% threshold following the previous study[81,82]. The reads were subsequently translated in six frames and scanned for *nirS*, *nirK*, *pmoA* with HiddenMarkovModels (HMM) using hmmsearch. HMM model was downloaded from the NcycFunGen database (https://zenodo.org/record/6636995)[83]. Sequence hits with an e-value cutoff score of $<10^{-5}$ were removed to ensure high confidence in all hits. OTUs defined as unclassified bacteria were reads without a recognizable match in the NcycFunGen database. These bacteria may represent novel or poorly characterized microbial taxa, requiring further investigation to determine their taxonomic classification and ecological significance.

**Shotgun metagenomic analyses.** Heavy DNA fractions from the $^{13}CH_4$-incubated samples were further used for shotgun sequencing to reveal the metabolic pathways associated with the coupling between soil $CH_4$ oxidation and denitrification. Paired-end fragment libraries with the insert size of 500 bp were constructed using NEXTFLEX Rapid DNA-Seq (Bioo Scientific, Austin, TX, USA). Adapter-appended fragments were sequenced on Illumina NovaSeq platform by Merjorbio (Shanghai, China). Raw sequencing reads were trimmed to low-quality reads and N-containing reads by the fastp software (https://github.com/OpenGene/fastp, v0.19.6)[84]. As a result, a total of 10,162.46–18,720.82 Mb (10.16–18.72 Gb) clean reads were kept, accounting for 92.6%–97.8% of raw bases. Then, the qualified reads were individually assembled into contigs using Megahit (https://github.com/voutcn/megahit, -k-min 47 -k-max 97 -k-step 10, v1.2.9), and Prodigal was used to predict open reading frames based on assembly results[85]. The assembly lengths for the assembly statistics were 444.10–811.21 Mb, comprising 621,075–1,206,876 contigs in total. Metabat2 (https://bitbucket.org/berkeleylab/metabat, v 2.12.1) was used to bin the contigs assembled from three replicates, specifically targeting those with lengths exceeding 1000 bp[86]. DAS_tools (https://github.com/cmks/DAS_Tool, v1.1.0) and CheckM (https://github.com/Ecogeno-mics/CheckM/wiki, v1.1.6) were used to respectively dereplicate and evaluate all recovered bins to obtain non-redundant high-quality draft genomes[87,88]. The coverage of MAGs (%) in each sample was estimated using CoverM (v0.6.1), and 21.28% to 49.07% of the reads were successfully mapped back to the assemblies. The relative abundance of each MAG is calculated using the Quant_bins module in MetaWRAP (v1.3)[89]. Taxonomic classification of MAGs was performed using GTDB-Tk v0.3.3 classify_wf command against the GTDB (http://gtdb.ecogenomic.org/, v2.3.2)[90,91]. An overview of the assembly statistics for MAGs that affiliated with methanotrophs and denitrifiers in the three selected soils was provided in Supplementary Table 2. Among these MAGs, 30 were classified as medium-quality according to the Minimum Information about a Single Amplified Genome (MIMAG; >50% completeness, <10% contamination)[92]. Only fairly complete MAGs with more over 80% completeness and less than 10% contamination were selected for further analysis[93–96]. To reveal the novelty of MAGs, fastANI (v1.0) was used to calculate the values of genome ANI as referred to GTDB database[97]. Genes from different samples were combined and clustered using CD-HIT (http://www.bioinformatics.org/cd-hit/) to remove redundant sequences (sequence identity

threshold 90% and alignment coverage threshold 90%)[98]. High-quality reads were aligned to the non-redundant gene catalogs to calculate gene abundance with 95% identity using SOAPaligner (v2.04)[99]. The predicted genes of MAGs were annotated using the HMM profile database for KEGG orthology with predefined score thresholds through KofamScan (v1.3.0)[100]. Specifically for denitrification genes, we utilized HMM-based search tools against the curated database from the previous study[101]. Transcripts per Million (TPM) values were calculated to reflect the relative abundance of genes in MAGs. The log (TPM + 1) values were calculated for visualization of the relative abundance data[102–104].

**$^{13}$C-metabolomics.** We further conducted $^{13}$C-metabolomics analyses to identify the intermediates derived from the oxidation of $^{13}CH_4$ that support denitrification. The experiments were conducted with three treatments, including no addition of $CH_4$ as control (CT), the addition of 10% $^{13}CH_4$ ($^{13}CH_4$), and the simultaneous addition of 10% $^{13}CH_4$ and 100 mg kg$^{-1}$ $NO_3^-$-N ($^{13}CH_4$ _N). By adding $NO_3^-$-N, we aimed to investigate how the intermediates derived from $CH_4$ oxidation change with the stimulation of denitrification activities. Indeed, there were significant increases in the transcription of denitrification genes including *norB*, *nosZI* and *nosZII* genes under $NO_3^-$-N addition (Supplementary Fig. 7). The cultures were comprised of 10.0 g of soil, 15 ml water and 0.01 mg DMPP. All microcosms were incubated at 25 °C in the dark for 15 days. At the end of incubation, soil samples were quickly frozen with liquid nitrogen and stored at −80 °C in refrigerator for metabolite extraction. Metabolites were extracted from 1 g soil using 400 μl of methanol-water (4:1, v/v) solution. The mixture was allowed to settle at −20 °C and then processed with a TissueLyser (JX-24, Jingxin, Shanghai, China) with beads at 40 Hz for 4 min at 50 Hz for 6 min, followed by vortexing for 30 s and ultrasonication at 40 kHz for 30 min at 5 °C. The samples were then maintained at −20 °C for 30 min to precipitate proteins. After centrifugation at $13,000 \times g$ at 4 °C for 15 min, the supernatant was carefully transferred to sample vials for metabolite profiling. Meanwhile, a pooled quality control sample (QC) was prepared by mixing equal volumes of all samples. The QC samples were disposed and tested in the same procedure as the analytic samples. All the analytical standards and internal standards were prepared individually at the concentration of 1 mg ml$^{-1}$ as stock solution. A standard working solution of 5 μg ml$^{-1}$ for each standard in 50% methanol was prepared by mixing each standard stock solution (1 mg ml$^{-1}$). The samples of calibration curves were finally prepared by isometrically mixing the serially diluted standard solution with internal standards solution to generate calibration levels covering a range of 1–2500 ng ml$^{-1}$. Both LC-MS/MS and GC-MS experiments were conducted at ProfLeader in Shanghai, China to determine a series of $^{13}$C-carbonaceous organics involved in $CH_4$ oxidation, including formaldehyde, formate, short-chain fatty acids, and intermediates involved in the serine cycle, gluconeogenesis, pyruvate metabolism and TCA cycle (Supplementary Table 3).

The LC-MS/MS experiments were carried out on an Agilent 1290 Infinity II UHPLC system coupled to a 6470A Triple Quadrupole mass spectrometry (Santa Clara, CA, United States). Samples were injected into a BEH C18 column (100 mm × 2.1 mm, 1.7 μm) at a flow rate of 0.35 ml min$^{-1}$. The mobile phase consisted of (A) water with 0.1% formic acid and (B) acetonitrile with 0.1% formic acid. The chromatographic separation was conducted by a gradient elution program as follows: 0–1 min, 5% B; 1.5–4.5 min, 25% B; 9 min, 50% B; 10–11.3 min, 100% B; 11.4–13 min, 5% B. The eluted analysts were ionized in an electro spray ionization source in negative mode (ESI-). The temperatures of source drying gas and sheath gas were 300 °C. The flow rates of source drying gas and sheath gas were 5 and 11 l min$^{-1}$, respectively. The pressure of nebulizer was 45 psi, and capillary voltage was 2500 V. All metabolite identification and isotopic enrichment were determined by MassHunter Workstation Software (version B.08.00, Agilent) using

the default parameters and assisting manual inspection. Corrections for natural abundance of $^{13}C$ were conducted on the $^{13}C$-labeled fractions[105].

GC-MS experiments were performed using an Agilent 7890/ 5975 C GC-MS (Agilent Corp, Santa Clara, CA, USA). The column was a DB-23 fused-silica capillary column (20 m × 0.18 mm × 0.2 μm). High pure helium (>99.999%) was used as a carrier gas at a constant flow rate of 1 ml min$^{-1}$ through the column. Injection volume was 1.5 μl in splitless mode, and the solvent delay time was 5.6 min. The initial oven temperature was held at 50 °C for 0.5 min, ramped to 170 °C at a rate of 15 °C min$^{-1}$, to 210 °C at a rate of 20 °C min$^{-1}$, to 240 °C at a rate of 15 °C min$^{-1}$ and held for 2 min. The temperatures of injector, transfer line, and electron impact ion source were set to 250 °C, 250 °C, and 230 °C, respectively. The impact energy was 70 eV, and data was collected in a full scan mode ($m/z$ 50–600). The raw data were processed by ChemStation Software (version E.02.02.1431, Agilent) by using the default parameters and assisting manual inspection to ensure the qualitative and quantitative accuracies of each compound. The correction of natural isotope abundance and MID were performed according to previous study[105].

### Statistical analysis

Pearson correlation was used to determine the associations between denitrification rate and $CH_4$-oxidizing activity and their functional gene abundances in R 4.0.2. Moreover, soil denitrification rate, $CH_4$-oxidizing activity, and gene copies were normalized using microbial biomass C (i.e., gene copies per unit microbial biomass C) to reflect the spatial interlinkages regardless of the total abundance of microbial biomass in soils[106]. Furthermore, the effects of climatic factors (mean annual precipitation and mean annual temperature), soil properties (pH, SOC, ammonia, $NO_3^-$), bacterial community composition (NMDS), gene abundances (nirK, nirS, pmoA) and $CH_4$-oxidizing activity on denitrification rate were evaluated by SEM. A priori model was established based on our current knowledge of the impacts of environmental factors on denitrification rate (Supplementary Fig. 4). The chi squared test ($\chi^2$; the model has a good fit when $0 \leq \chi^2/d.o.f \leq 2$ and $0.05 < p \leq 1.00$) and the root mean square error of approximation (RMSEA, the model has a good fit when RMSEA is indistinguishable from 0) were used to test the overall goodness of fits for the model[107,108]. SEM analyses were conducted using AMOS 21.0 (spsss Inc., Chicago, IL, USA). Climatic variables were collected from the Worldclim database (https://www.worldclim.org) using ArcGIS V10.6 software.

Next, we established a co-occurrence network to identify the potential associations between denitrifiers and aerobic methanotrophs in the field surveys. The denitrifiers were shown in Supplementary Table 5. We used the psych package of R 4.0.2 to make spearman correlation matrix for the known denitrifiers and methanotrophs at the family level. We considered a co-occurrence to be robust only if the spearman's correlation coefficient was >0.50 and $p < 0.05$. The network was visualized using Gephi 0.9.2[109]. Furthermore, soil denitrification rate was normalized by the corresponding $CH_4$-oxidizing activity to reflect the relative denitrification to $CH_4$ oxidation activities. We then used random forest analysis to identify significant ($p < 0.05$) environmental predictors of the $CH_4$-oxidizing activity-normalized denitrification rate, with higher MSE% indicating more important variables.

In the microcosm experiments, one-way analysis of variance (ANOVA) was used to detect the significant differences between treatments by SPSS Statistics 21.0 (IBM, Chicago, IL). LEfSe analysis ($p < 0.05$, LDA score >2) was used to identify bacterial biomarkers for two groups on the open website (http://huttenhower.sph.harvard.edu/galaxy). We used DESeq2 (v1.36.0) to identify OTUs significantly enriched in the heavy fractions of the $^{13}C$-labeled treatments compared to the heavy fractions in the corresponding $^{12}C$ controls[110]. Incorporators

were defined as OTUs with log$_2$-fold change values higher than 1 with adjusted $p$ values (FDR-adjusted $p$-value) lower than 0.05. The relative abundance ($Z$-score standardized) of the top 25 genera within the $^{13}C$-heavy fractions was visualized via the pheatmap package.

### Reporting summary

Further information on research design is available in the Nature Portfolio Reporting Summary linked to this article.

## Data availability

The gene amplicon sequences and metagenomic sequences generated in this study have been deposited to the NCBI SRA database under the BioProject IDs of PRJNA1096118, PRJNA1097355 and PRJNA1096656. The databases used in this study include Worldclim database (https://www.worldclim.org), SILVA 128 (https://www.arb-silva.de/documentation/release-128/), NcycFunGen database (https://zenodo.org/record/6636995), Genome Taxonomy Database (http://gtdb.ecogenomic.org/) and KEGG database (https://ww.genome.ad.jp/kegg/). Additional figures and tables can be found in the Supporting Information. Source data are provided with this paper.

## Code availability

All code associated with our analyses in this study is available at https://figshare.com/s/c1a8ad171646eae9b6c9; https://doi.org/10.6084/m9.figshare.22631041.

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

## Acknowledgements

W.T. was supported by the National Key Research and Development Program of China (2021YFD1901205). J.F. was supported by the National Natural Science Foundation of China (32071595 and 32371731). Y.-R.L. was supported by the National Natural Science Foundation of China (42177022). P.C. was supported by the Fundamental Research Funds for the Central Universities (2662023PY010). M.D.-B. was supported by a project from the Spanish Ministry of Science and Innovation (PID2020-115813RA-I00), and a project of the Fondo Europeo de Desarrollo Regional (FEDER) and the Consejería de Transformación Económica, Industria, Conocimiento y Universidades of the Junta de Andalucía (FEDER Andalucía 2014-2020 Objetivo temático "01-Refuerzo de la investigación, el desarrollo tecnológico y la innovación") associated with the research project P20_00879 (ANDABIOMA).

## Author contributions

Y.-R.L. and J.F. designed the study. K.-H.C. carried out the main experiments and analyzed the data with help from Y.-R.L. and J.F. K.-H.C. and J.F. wrote the paper. Y.-R.L., J.F., P.L.B., Z.Y., Q.H., M.D.-B., P.C. and W.T. revised the paper. All authors reviewed the paper and approved the final version of the manuscript.

## Competing interests

The authors declare no competing interests.
