## [Peer Review File · Nature Communications]

Reviewers' Comments:

Reviewer #1:

Remarks to the Author:

The manuscript by Chen et al. investigates the impacts of methane oxidation on denitrification in flooded croplands. It is a well-planned and well-executed study. Three pieces of evidence are presented:

A correlation between methane oxidation and N₂O emissions in hypoxic soils supplemented with nitrate fertilizers.

Expression of denitrification pathways under methane supplementation.

Co-occurrences of methane-consuming bacteria and microbial species known for their denitrifying capabilities.

Overall, the coupling between methane oxidation and N₂O production is well supported. However, there are a few elements of the study that would benefit from additional clarifications:

It could be expected that methane oxidation removes oxygen, and hypoxic conditions stimulate denitrification, and thus the observed correlations might not be direct. The study provides no metabolic evidence that methane-derived organic compounds enhance denitrification. Therefore, no data could support "mutualistic" interactions but rather commensalism. I encourage the author to investigate metabolites (¹³C-carbon), if still possible, or look deeper into gene expression studies (see comment below).

Most of the observed denitrifiers are known for their C₁-capabilities (methanol/ formate utilization); the microbes also prefer to use acetate. While metagenomic evidences are presented, it is hard to find a microorganism that does not possess formaldehyde detoxification or formate oxidation pathways. Changes in the gene expression that could be connected to C-metabolism in denitrifiers were not presented (i.e., changes in gene expression for C₁/C₂ acid assimilation/oxidation or methanol oxidation).

The ability of methanotrophs themselves to produce N₂O needs to be better described.

Ideally, a series of methanogenic microcosms should be set to provide information on the in-situ flux of methane, i.e., specific to conditions studies (see my comments on microcosms with 10% methane).

Only a subset of functions related to denitrification was studied.

Minor comments:

L166. All genes should be defined by their function. For example, FdhA can be assumed to be a formate dehydrogenase, rarely present in methanotrophic bacteria.

L171. Please check fae (i.e., formaldehyde activating enzyme), not hydrolyse.

L415. 10% methane is very high. Please justify; provide additional data supporting such a high production rate for the studied soils (i.e., SOC).

Fig 5 Should be corrected to reflect the fae function. No data for the H₄MTP pathway were presented. The metabolic entry to the serine cycle is methylene-H₄folate; thus, some parts of the C₁-metabolism are missing. Some direct measurements should be presented to support formaldehyde/methanol/formate secretion. Additional metabolic (or at least in-silico) support for acetate production from the serine cycle should be added, as acetyl-CoA can also be produced via phosphoketolase and pyruvate dehydrogenase.

Reviewer #2:

Remarks to the Author:

Manuscript Number: NCOMMS-23-14274

Title: Species interactions and metabolic couplings between soil aerobic methanotrophs and denitrifiers in flooded croplands

The authors investigated the role of aerobic CH₄ oxidation combined with denitrification in flooded croplands. Then, ¹³CH₄-DNA-stable isotope probing experiments and microcosm experiments were carried out to confirm the co-existence and species interactions between aerobic CH₄ oxidation and denitrification. They claim that for the first time identify key microbial taxa and pathways driving the coupling between aerobic CH₄ oxidation and denitrification in flooded soils. After reading carefully this manuscript, considering the mechanism explanation and the innovation of the article, the quality of this work was not enough for high standard of Nature Communications.

Major comments

1. The coupling relationship between aerobic methane oxidation and denitrification was studied, and N₂O release was measured. However, only the abundance of nirK/nirS cannot fully represent the whole denitrification process. First of all, it is not clear whether NO₃⁻ or NO₂⁻ is the substrate of aerobic methane oxidation coupled denitrification; Secondly, the q-PCR results of functional genes in the denitrification process were incomplete, especially the transcription level of functional genes involving NO₂⁻ to NO and NO to N₂O were not explained clearly.
2. The author enumerates several intermediate carbonaceous organics released by aerobic methane oxidation, and believes that they can be used as the carbon source of its coupled denitrification process. (P162-167) However, the carbon in the soil itself can also support the occurrence of denitrification. How can the author distinguish whether it is the carbon in the soil itself or the intermediate carbonaceous organics that drives the denitrification process?
3. In addition to denitrification, N₂O is also released during ammonia oxidation, especially under aerobic conditions. Therefore, the N₂O released by the system should be a comprehensive result, which does not fully represent the N₂O released during the denitrification process. In this study, the author used 3,4-dimethylpyrazole photosphate to inhibit the nitrification process. How can we ensure complete inhibition? The author should provide the results of the inhibition experiment. P114-123 ; P383-384
4. Why does aerobic methane oxidation occur in flooded fields? Which is the main process for aerobic or anaerobic methane oxidation? P225-227
5. Accurate denitrification activity should be measured using the ¹⁵N isotope technique. P293-294

Minor comments

1. Where is the Fig 1A? P98-103
2. The physical and chemical properties of black soil, red soil, and yellow soil in the text are missing.
3. Three soil collection locations are missing.
4. Why do DEA and MOA represent the activity of denitrification and aerobic methane oxidation, respectively?

Reviewer #3:

Remarks to the Author:

All in all, I think this study has value. However, I think the manuscript needs serious modifications. As written, the results are too brief and the discussion is too long and very repetitive. Most important, the authors somehow suggest that they discovered the connection between aerobic methane oxidation and denitrification while they did not. What is valuable about their work is the presentation of a transect of the data across soils in China that is very useful, while the concept itself existed for quite a while, needing more data/proof. Neither they discovered any novel pathways, these pathways have been known for decades and even centuries. So I think the manuscript should be rewritten to clearly state the goal of providing a solid proof for something that has been known for quite a while. I suggest that you expand the Results section to state the goals you tried to pursue with every experiment you conducted. How each experiment leads to the next experiment. In Discussion, provide a brief summary of what is new that you discovered. In my opinion, you present a bigger study on what has been known, but it is important. Please pay more attention to figure legends and expand. For example, what are the symbols a, b, ab etc. stand for? What is the role of MDO in methylotrophy? Please explain what it does in the process that you suggest. Can you be more specific about which organics might be driving the denitrification process. You mention multicarbon substrates in some instances and monocarbon substrates in others. How would Rhizobia use methanol? If they are methylotrophic do you have proof for that?

Response letter

Responses to reviewer's comments:

Reviewer #1:

The manuscript by Chen et al. investigates the impacts of methane oxidation on denitrification in flooded croplands. It is a well-planned and well-executed study. Three pieces of evidence are presented:

A correlation between methane oxidation and N₂O emissions in hypoxic soils supplemented with nitrate fertilizers.

Expression of denitrification pathways under methane supplementation.

Co-occurrences of methane-consuming bacteria and microbial species known for their denitrifying capabilities.

Overall, the coupling between methane oxidation and N₂O production is well supported. However, there are a few elements of the study that would benefit from additional clarifications.

1. Thank you very much for your positive comments on our manuscript. We have carefully addressed all your concerns regarding clarifications on the experimental setup and the interpretation of results. Additionally, we conducted a series of experiments including reconstruction of metagenome-assembled genomes (MAGs) and utilization of ¹³C-metabolites to investigate the metabolic process underlying the coupling of aerobic methane (CH₄) oxidation and denitrification. We now provide our point-to-point responses to your concerns as follows.

It could be expected that methane oxidation removes oxygen, and hypoxic conditions stimulate denitrification, and thus the observed correlations might not be direct. The study provides no metabolic evidence that methane-derived organic compounds enhance denitrification. Therefore, no data could support "mutualistic" interactions but rather commensalism. I encourage the author to investigate metabolites (¹³C-carbon), if still possible, or look deeper into gene expression studies (see comment below).

2. Thanks for these constructive comments. We agree that denitrifiers may benefit from oxygen depletion resulted from CH₄ oxidation. We have discussed this important point in lines 308-325. To further substantiate the mutualistic interactions between microbial taxa involved in CH₄ oxidation and denitrification, we reconstructed MAGs using ¹³C-metagenomic data and conducted additional ¹³C-metabolomics experiments. Results indicated that essential denitrifiers, such as Species Genome Bins (SGBs) belonged to the genera *Magnetospirillum* and *Novimethylophilus*, significantly enriched genes including *acs*, *pct*, *ato*, *fae*, and *fdo*. These enrichments of C-utilization genes indicate that the denitrifiers possess the metabolic potential to utilize intermediates released during CH₄ oxidation, including short-chain fatty acids (e.g., acetate, propionate, and butyrate), organic

acids involved in pyruvate metabolism and TCA cycle (e.g., lactate, malate, and succinate), formaldehyde, and formate (Fig. 5A; Fig. S14). Moreover, none of these identified denitrifiers were found to carry *pmo* and *mmo* genes (encoded soluble CH₄ monooxygenase), providing additional support for the notion that these denitrifiers probably depend on intermediates released from methanotrophs, rather than directly utilizing CH₄.

Furthermore, our ¹³C-metabolomics results confirmed that intermediates such as acetate, propionate, butyrate, and lactate derived from ¹³CH₄ oxidation could serve as C sources for denitrifiers (Fig. 5B; Figs. S15-16). Compared with ¹³CH₄ addition, simultaneous addition of ¹³CH₄ and nitrate (NO₃⁻) induced an overall higher ¹³C-labeled fraction of acetate, propionate, butyrate and lactate, but a lower concentration of these intermediates in the black and yellow soils. These results suggest an increased consumption of these intermediates in response to the stimulation of denitrification activities through the addition of NO₃⁻. We have incorporated this evidence of mutualistic interactions in Figs. 5, S14, S15-16 and in lines 192-252 and 340-390 in the revised manuscript.

Fig. 5 The proposed metabolic pathways of the coupling between aerobic methane (CH₄) oxidation and denitrification in the flooded soils. **A** The color gradients represent the counts of major genes in the corresponding metagenome-assembled genomes (MAGs) classified as methanotrophs and denitrifiers, respectively, in the heavy DNA from ¹³CH₄ incubation. The counts of genes were log (n+1) transformed. The definitions of genes are listed in Table S7. **B** Changes in the ¹³C-labeled fraction of metabolites derived from ¹³CH₄ oxidation in the three typical soils. The error bar represents the standard error of triplicate samples (n = 3). Different lowercase letters in B indicate significant differences between treatments (*p* < 0.05).

Most of the observed denitrifiers are known for their C1-capabilities (methanol/ formate utilization); the microbes also prefer to use acetate. While metagenomic evidences are presented, it is hard to find a microorganism that does not possess formaldehyde detoxification or formate oxidation pathways. Changes in the gene expression that could be connected to C-metabolism in denitrifiers were not presented (i.e., changes in gene expression for C1/C2 acid assimilation/oxidation or methanol oxidation).

3. We agree with the reviewer that many denitrifiers have the capacity to utilize C1 and C2 organics. The pivotal aspect is to demonstrate whether these C1 and C2 organics originate from CH₄ oxidation, thereby establishing the metabolic coupling between CH₄ oxidation and denitrification within intricate soil systems. To address this concern, we conducted ¹³CH₄-DNA-stable isotope probing (SIP) experiments to identify denitrifiers that assimilate C derived from ¹³C-labeled CH₄. Additionally, we reconstructed MAGs using ¹³C-metagenomic data to explore enriched genes involved in carbonaceous organics in denitrifiers following the reviewer's suggestions. Our ¹³C-MAG analyses showed significant enrichments of genes associated with formaldehyde (*fae*), formate (*fdo*, *fdsD*, *fdw*, *fdh*), acetate (*acs*), propionate (*pct*), butyrate (*ato*), lactate (*ldh*), malate (*fum*) and succinate (*suc*) utilization in denitrifiers (Fig. 5A; Fig. S14). These results provide crucial insights into the metabolic pathways of aerobic CH₄ oxidation and denitrification in the soil.

Also, we conducted additional ¹³C-metabolomics experiments to further reveal the metabolic pathways of the coupling between CH₄ oxidation and denitrification. The ¹³C-metabolomics results confirm the results of ¹³C-MAG analyses, and indicate that aerobic CH₄ oxidation could couple with denitrification through intermediates including short-chain fatty acids (e.g., acetate, propionate, and butyrate) and organic acids involved in pyruvate metabolism and TCA cycle (e.g., lactate, malate, and succinate). Specifically, ¹³C-metabolomics results showed that the simultaneous addition of ¹³CH₄ and NO₃⁻ induced an overall higher ¹³C-labeled fraction of acetate, propionate, butyrate, and lactate compared with ¹³CH₄ addition in the black and yellow soils (Fig. 5B; Figs. S15-16). Moreover, the concentration of these intermediates was generally lower under the simultaneous addition of ¹³CH₄ and NO₃⁻, suggesting increased consumption of CH₄-derived intermediates in response to the stimulation of denitrification activities under NO₃⁻ addition. Please see our revisions in Figs. 5, S14-16, and in lines 192-252 and 340-390.

The ability of methanotrophs themselves to produce N₂O needs to be better described.

4. Thanks for this valuable suggestion. We have discussed the capacity of methanotrophs themselves in producing N₂O as suggested (Lines 374-381). We noticed that some methanotrophs possess homologues of denitrification genes, such as *nirS*, *nirK*, *norB*¹. Although we found the presence of partial denitrification genes (including *nar*, *nir*, and *nor* genes) in several SGBs of methanotrophs through the analysis of MAGs, others lack one or all of these genes (Figs. 5A, S13-14). Additionally, previous studies have indicated that while some microbial strains may carry denitrification genes, it remains necessary to verify whether these genes are expressed and capable of facilitating denitrification^{2,3}. Therefore, extensive experiments are warranted to fully comprehend the role of methanotrophs in driving denitrification. Moreover, it is noteworthy that the MAG analyses of methanotrophs did not identify homologs of *nosZ* gene, consistent with the results from previous studies^{4,5}. Therefore, the elevated expression of *nosZ* gene under the addition of CH₄ or methanotrophs suggests that it is an important process for CH₄ oxidation to affect denitrification by influencing denitrifiers, not only through the expression of denitrification genes within methanotrophs themselves. Therefore, we have incorporated statements to clarify the influences of aerobic methanotrophs on denitrification, both through carrying denitrification genes themselves and indirectly through their effects on denitrifiers in the revised manuscript (Lines 207-211 and 374-381).

Ideally, a series of methanogenic microcosms should be set to provide information on the in-situ flux of methane, i.e., specific to conditions studies (see my comments on microcosms with 10% methane).

5. We understand your concerns. We agree that a series of methanogenic microcosms could better reflect in-situ CH₄ emissions. However, we also recognize that significant variations in CH₄ fluxes occur across different sites because of differences in soil properties, management practices, climatic conditions, etc⁶. Therefore, it is difficult to accurately and simultaneously simulate in-situ CH₄ emission fluxes, especially at a large scale across diverse climatic regions. Previous studies have indicated that wetland environments, such as rice paddies, often exhibit substantial CH₄ production due to hypoxic conditions and high levels of organic matter accumulations⁷. In this study, soil organic carbon (SOC) content was as high as 37.53 g kg⁻¹ in Northeast China soils (Table S1), providing an ample carbon source for CH₄ production. Within these high SOC and hypoxic systems, previous studies indicated that monthly in-situ porewater dissolved CH₄ concentrations in mud and water-covered soils exceeded 0.15 mM L⁻¹^{7,8}. This CH₄ concentration in the pore water could be obtained by an initial CH₄ concentration of 10% (v/v) in the air, calculated by Henry's Law and the van't Hoff equation to account for temperature-dependent solubility of CH₄⁹. Additionally, considering the relatively high cost of DNA-SIP experiments and the need for sufficient ¹³C markers to enhance the accuracy of identification of key microbial taxa and metabolic pathways involved, we used 10% CH₄ concentration (v/v) following previous

DNA-SIP experiments in flooded soils¹⁰⁻¹². We have clarified this important point in lines 308-317 and 561-574 of the revised manuscript.

Only a subset of functions related to denitrification was studied.

6. Thanks for this constructive comment. We have carried out additional experiments to examine the consumption of NO_3^- and the abundances of more denitrification genes in the CH_4 and methanotrophs addition experiments following your suggestions. The supplementary genes included those encoding the reductase of NO_3^- (*narG*), nitric oxide (*norB*) and nitrous oxide (*nosZI*, *nosZII*) (Fig. S7; Fig S10). Overall, the addition of both CH_4 and methanotrophs generally enhanced the transcription of these supplementary genes, either partially or entirely. We have added the associated Methods and Results in lines 131-144, 145-160 and 295-304, and Figs. S7; S10.

Minor comments:

L166. All genes should be defined by their function. For example, *FdhA* can be assumed to be a formate dehydrogenase, rarely present in methanotrophic bacteria.

7. We have defined the functions of genes throughout the manuscript such as in lines 198-207. We also double-checked the presence of genes and potential pathways of methanotrophic bacteria by reconstructing MAGs, and did not detect the presence of *FdhA*. Therefore, we have removed statements associated with *FdhA* from the manuscript.

L171. Please check *fae* (i.e., formaldehyde activating enzyme), not hydrolyse.

8. Checked and corrected in Line 201.

L415. 10% methane is very high. Please justify; provide additional data supporting such a high production rate for the studied soils (i.e., SOC).

9. We appreciate your concern. As pointed out by the reviewer, there were substantial accumulations of SOC in anaerobic paddy fields. The SOC concentration was as high as 37.53 g kg^{-1} in this study, providing adequate substrates for methanogenesis and heterotrophic microorganisms. Previous studies have indicated that monthly in-situ porewater dissolved CH_4 concentrations in mud and water-covered soils were higher than 0.15 mM L^{-1} ^{7,8}. As responded above, this CH_4 concentration in the pore water could be obtained by an initial CH_4 concentration of 10% (v/v) in the air. Additionally, to ensure accurate identification of key microbial taxa and metabolic pathways, we used 10% CH_4 concentration (v/v) following previous ^{13}C - CH_4 -SIP experiments in flooded croplands¹⁰⁻¹². We have incorporated supplemental data in Table S1, and made corresponding clarifications in lines 308-317 and 561-574 of the revised manuscript.

Fig 5 Should be corrected to reflect the *fae* function. No data for the H₄MTP pathway were presented. The metabolic entry to the serine cycle is methylene-H₄folate; thus, some parts of the C1-metabolism are missing. Some direct measurements should be

presented to support formaldehyde/ methanol/ formate secretion. Additional metabolic (or at least in-silico) support for acetate production from the serine cycle should be added, as acetyl-CoA can also be produced via phosphoketolase and pyruvate dehydrogenase.

10. Thanks for the suggestion. We have corrected the definition and function of the *fae* gene as “formaldehyde activating enzyme” (line 201; Fig. 5A). Additionally, we have supplemented the metabolic pathways of CH₄ oxidation following the results of MAGs, including H₄MTP pathway, methylene-tetrahydrofolate (H₄F)-linked C₁ transfer pathway, as well as pathways associated with acetate production and serine cycle. Please see additions in Fig. 5A. Moreover, we conducted additional ¹³C-metabolomics experiments to provide more direct evidence to support the secretion of the metabolites from CH₄ oxidation (Fig. 5B; Figs. S15-16). In particular, we observed an overall increase in the ¹³C-labeled fraction of acetate, propionate, butyrate and lactate together with a simultaneous decrease in the concentration of these intermediates under combined ¹³CH₄ and NO₃⁻ addition. This observation suggests a significant consumption of these intermediates in response to the stimulation of denitrification activities under NO₃⁻ addition. These results imply that short-chain fatty acids (e.g., acetate, propionate, and butyrate) and organic acids involved in pyruvate metabolism and TCA cycle (e.g., lactate), derived from ¹³CH₄ oxidation, may serve as the C source for denitrifiers. Please see our additions in lines 192-252 and 340-384 and Fig. 5.

Reviewer #2 (Remarks to the Author):

The authors investigated the role of aerobic CH₄ oxidation combined with denitrification in flooded croplands. Then, ¹³CH₄-DNA-stable isotope probing experiments and microcosm experiments were carried out to confirm the co-existence and species interactions between aerobic CH₄ oxidation and denitrification. They claim that for the first time identify key microbial taxa and pathways driving the coupling between aerobic CH₄ oxidation and denitrification in flooded soils. After reading carefully this manuscript, considering the mechanism explanation and the innovation of the article, the quality of this work was not enough for high standard of Nature Communications.

11. Thank you very much for your valuable feedback and comments. We have conducted additional experiments and made corresponding revisions to emphasize the novelty and improve the quality of our research. In particular, we conducted an array of additional experiments (e.g., ¹³C-metabolomics) to provide a more accurate representation of the underlying mechanisms involved in the coupling between aerobic methane (CH₄) oxidation and denitrification.

In brief, we have carried out the following additional experiments during revision: (1) evaluated the ¹³C-CH₄ metabolomics to offer direct evidence on the mutualistic interactions between soil aerobic methanotrophs and denitrifiers (Fig.

5B; Figs. S15-16); (2) reconstructed metagenome-assembled genomes (MAGs) to investigate changes in gene expression associated with C-metabolisms in denitrifiers (Fig. 5A; Fig. S14); (3) measured denitrification activity using the ^{15}N isotope technique (Fig. 1A; Fig. S1); (4) examined additional gene abundances, including *narG* encoding nitrate (NO_3^-) reductase, *norB* encoding nitric oxide (NO) reductase, *nosZI* and *nosZII* encoding nitrous oxide (N_2O) reductase, to provide a more comprehensive representation of the entire denitrification process (Figs. S7, S10); (5) verified the inhibitory effect of 3,4-dimethylpyrazole phosphate (DMPP) on nitrification (Fig. S19).

These experiments enable us to identify specific microbial taxa involved in aerobic CH_4 oxidation combined with denitrification in flooded paddy soils, expanding beyond the commonly studied pure culture systems. Additionally, we elucidated the metabolic pathways associated with this coupling by reconstructing ^{13}C -MAGs and measuring ^{13}C -metabolomics, providing new insights into the underlying mechanisms. These findings contribute to addressing a critical gap in the current research by providing compelling evidence on species and metabolic interactions between aerobic CH_4 oxidation and denitrification.

Our findings have important implications for improving the prediction of nitrogen fertilizer use efficiency and greenhouse gas (GHG) emission fluxes from flooded croplands, which are known hotspots for GHG emissions. Through a comprehensive large-scale survey across major rice-producing areas of China combined with systematic microcosm experiments, we have illustrated an effective approach that tackles an unexplored aspect of biochemical interactions in flooded croplands. We believe that the incorporation of our additional experiments and the highlights of these innovations significantly enhance the value of this article, aligning it with the high-quality requirements of Nature Communications. We sincerely appreciate your time and valuable feedback. Please find our point-to-point responses to your concerns below.

Major comments

1. The coupling relationship between aerobic methane oxidation and denitrification was studied, and N_2O release was measured. However, only the abundance of *nirK/nirS* cannot fully represent the whole denitrification process. First of all, it is not clear whether NO_3^- or NO_2^- is the substrate of aerobic methane oxidation coupled denitrification; Secondly, the q-PCR results of functional genes in the denitrification process were incomplete, especially the transcription level of functional genes involving NO_2^- to NO and NO to N_2O were not explained clearly.
12. Thanks for these constructive comments. We have conducted additional experiments to determine gene expression involved in the most denitrification processes, including *narG*, *norB*, *nosZI* and *nosZII* (Fig. S7; Fig. S10). Our findings reveal that the addition of both CH_4 and methanotrophs generally promoted the expression of denitrification genes encoding nitrite (NO_2^-) reductase (*nirK* and *nirS*) in the three representative soils, except for the medium and high concentrations of

CH₄ and methanotrophs addition in the red soil. These results indicate that NO₂⁻ could serve as the substrate for aerobic CH₄ oxidation-coupled denitrification in the representative soils.

However, the expressions of other denitrification genes varied in different soils. For instance, in the black and yellow paddy soils, the addition of CH₄ promoted the expression of *narG* and *nosZII*, suggesting that aerobic CH₄ oxidation could also be coupled with denitrification process utilizing NO₃⁻ and N₂O as substrates in these soils. In the red soil, however, the enhanced expression of *norB* and *nosZII* genes by addition of CH₄ and *narG* and *nosZI* genes by addition of methanotrophs indicates potential coupling of aerobic CH₄ oxidation with denitrification using NO₃⁻, NO and N₂O as substrates. These results highlight variations in the specific steps of the coupling between CH₄ oxidation and denitrification across different soils, thereby providing a more mechanistic understanding of this interaction of C and N cycles. We have incorporated these results in Figs. S7 and S10, and provided clarification regarding these findings in lines 131-144, 145-160 and 295-304 of the revised manuscript.

2. The author enumerates several intermediate carbonaceous organics released by aerobic methane oxidation, and believes that they can be used as the carbon source of its coupled denitrification process. (P162-167) However, the carbon in the soil itself can also support the occurrence of denitrification. How can the author distinguish whether it is the carbon in the soil itself or the intermediate carbonaceous organics that drives the denitrification process?

13. We acknowledge that carbonaceous organics themselves in the soil can contribute to denitrification process. Therefore, we conducted ¹³CH₄-isotope tracing experiments to distinguish whether the C-deriving denitrification process comes from the soil substrates or intermediates of CH₄ oxidation. Our ¹³CH₄-DNA-stable isotope probing (SIP) results showed significant enrichments of ¹³C in denitrifiers carrying the *nirK* and *nirS* genes. Through further new analysis of MAGs reconstructed from ¹³CH₄-SIP-metagenomics, we found that these denitrifiers enriched genes associated with formaldehyde (*fae*), formate (*fdo*, *fdsD*, *fdw*, *fdh*), acetate (*acs*), propionate (*pct*) and butyrate (*ato*), lactate (*ldh*), malate (*fum*), and succinate (*suc*) utilization (Fig. 5A; Fig. S14). As none of the identified denitrifying populations had the annotated potential for direct CH₄ utilization in our experiments, the ¹³C-enriched denitrifiers likely relied on ¹³CH₄-driven intermediates from methanotrophs. These ¹³CH₄-isotope tracing results provide crucial evidence of essential denitrifiers and metabolic pathways involved in the coupling between aerobic CH₄ oxidation and denitrification.

Furthermore, we conducted additional ¹³CH₄-metabolomics experiments to provide direct evidence of the metabolic coupling between CH₄ oxidation and denitrification (Fig. 5B; Figs. S15-16). We observed significant enrichments of ¹³C in acetate, propionate, butyrate, lactate, malate, and succinate under the combined

addition of $^{13}\text{CH}_4$ and NO_3^- compared with CH_4 amendment. The increased enrichments in ^{13}C , together with decreased concentrations of these intermediates, suggest enhanced consumption of these intermediates in response to the stimulation of denitrification activities under NO_3^- addition. The ^{13}C -metabolomics results validated the results of ^{13}C -MAG analyses, and indicate that these short-chain fatty acids and TCA cycle (e.g., acetate, propionate, and butyrate) and organic acids involved in pyruvate metabolism and TCA cycle (e.g., lactate, malate, and succinate) could serve as C sources for denitrifiers, thereby facilitating the coupling between CH_4 oxidation and denitrification. We have incorporated texts to clarify the purpose and results of these $^{13}\text{CH}_4$ experiments in Fig. S6 and in lines 192-252 and 340-384.

Figure S15. Typical LC-MS/MS and GC-MS chromatograms of targeted metabolites originating from methane (CH_4) oxidation in three typical soils. A

Short chain fatty acids (SCFAs); **B** The intermediates involved in pyruvate metabolism, TCA cycle, serine cycle and gluconeogenesis; **C** Formate; **D** Formaldehyde.

3. In addition to denitrification, N_2O is also released during ammonia oxidation, especially under aerobic conditions. Therefore, the N_2O released by the system should be a comprehensive result, which does not fully represent the N_2O released during the denitrification process. In this study, the author used 3,4-dimethylpyrazole phosphate to inhibit the nitrification process. How can we ensure complete inhibition? The author should provide the results of the inhibition experiment. P114-123; P383-384

14. We agree that the release of N_2O reflects the combined effects of both denitrification and nitrification processes. Therefore, we used DMPP to inhibit the nitrification process as done in many previous studies¹³⁻¹⁵. Our results clearly showed a significant 70.7-91.5% reduction in the nitrification process, as indicated by the diminished accumulation of NO_3^- and N_2O . These results indicate that N_2O emissions were predominantly (over 70%) attributed to denitrification with DMPP addition, although the inhibitor did not completely (100%) suppress ammonia oxidation. Therefore, we consider N_2O emissions with DMPP addition as a potential proxy for denitrification, following numerous previous studies¹⁶⁻¹⁸. We have thus toned down the language to consider N_2O production as a representative of potential denitrification activities in our microcosm experiments (lines 517-524). Furthermore, we determined denitrification activity using ^{15}N isotope technique, and conducted additional denitrification functions such as NO_3^- consumption and more gene transcription analysis, to substantiate our results regarding N_2O production through denitrification. Please see our additions in Figs. S6-7, S10, S19 and lines 126-160 and 529-533.

4. Why does aerobic methane oxidation occur in flooded fields? Which is the main process for aerobic or anaerobic methane oxidation? P225-227

15. Rice plants, like other aquatic plants, possess a gas vascular system, which allows the diffusion of oxygen (O_2) to the roots to support respiration. A portion of the O_2 leaks from the roots and creates a shallow oxic zone. Consequently, flooded paddy fields are structured ecosystems where the surface layer is partially oxic or hypoxic, supporting the coexistence of various aerobic and anaerobic microbes¹⁹. Notably, previous studies have indicated that these surface layers provide a suitable habitat for the activity and proliferation of aerobic methanotrophs, which are responsible for consuming over 70% of the produced CH_4 before escaping into the atmosphere^{20,21}. Consistently, our field survey in the surface layer and microcosm experiments did not reveal the presence of anaerobic methanotrophic archaea and bacteria (e.g., NC10 bacteria), which may be due to that the surface layer of paddy soil was not a preferred habitat for these taxa. Therefore, we propose that aerobic CH_4 oxidation could be the main process of CH_4 consumption in flooded soils. We incorporated texts to explicate the importance of aerobic CH_4 oxidation in lines 55-60 and 308-325.

5. Accurate denitrification activity should be measured using the ^{15}N isotope technique. P293-294

16. Thanks for this valuable comment. We have now re-measured denitrification activity using the ^{15}N isotope technique in the revised manuscript following your suggestion. The denitrification activity obtained using ^{15}N isotope technique did not alter the overall patterns observed across different climatic regions, nor did it influence the relationships with CH_4 oxidation activity. We have provided detailed information regarding these measurements in lines 107-109 and 463-475.

Minor comments

1. Where is the Fig 1A? P98-103

17. We have cited Fig. 1A and included the corresponding statements in the results. “Specifically, denitrification rates (DR) and CH_4 -oxidizing activities (MOA) ranged from 0.78 to 164.48 $\text{nmol } ^{15}\text{N g soil}^{-1} \text{ h}^{-1}$ and 0.31 to 38.81 $\mu\text{g CH}_4 \text{ g}^{-1} \text{ soil} \cdot \text{h}^{-1}$, respectively (Fig. 1A).” (Lines 107-109).

2. The physical and chemical properties of black soil, red soil, and yellow soil in the text are missing.

18. Provided as suggested in Table S1 and lines 126-129.

3. Three soil collection locations are missing.

19. Added as suggested in Table S1.

4. Why do DEA and MOA represent the activity of denitrification and aerobic methane oxidation, respectively?

20. Denitrification enzyme activity quantified the production of N_2O resulting from the reduction of potassium nitrate (KNO_3), and is commonly regarded as a surrogate of denitrification rate²². To obtain accurate determination of denitrification activity, we determined denitrification rates using the ^{15}N isotope technique following your comments. The activity of CH_4 oxidation is often determined by measuring the consumption of CH_4 over a specific period^{23,24}, encompassing both aerobic and anaerobic CH_4 oxidation in flooded soils. We have corrected the term “aerobic CH_4 oxidation activity” to “ CH_4 oxidation activity” to prevent potential misinterpretation. In this study, the anaerobic methanotrophic archaea and bacteria (e.g., *Methylomirabilis oxyfera*-like NC10 bacteria) were not detected in the surface layer of flooded paddy fields in this study. This is consistent with previous studies that over 70% of CH_4 was consumed by aerobic methanotrophs before escaping into the atmosphere^{20,21}. Therefore, we have carefully discussed that the observed CH_4 oxidation may be predominantly a result of aerobic CH_4 oxidation in lines 308-325. Thanks again for all the constructive comments.

Reviewer #3 (Remarks to the Author):

All in all, I think this study has value. However, I think the manuscript needs serious modifications. As written, the results are too brief and the discussion is too long and very repetitive.

21. Thank you very much for your positive comments regarding the value of this study. We have made thorough revisions to modify the manuscript. In particular, we have carefully reviewed the results section to incorporate more detailed information and have addressed all the concerns related to the length and repetition in the discussion section. Please see our point-to-point responses below.

Most important, the authors somehow suggest that they discovered the connection between aerobic methane oxidation and denitrification while they did not. What is valuable about their work is the presentation of a transect of the data across soils in China that is very useful, while the concept itself existed for quite a while, needing more data/proof. Neither they discovered any novel pathways, these pathways have been known for decades and even centuries. So I think the manuscript should be rewritten to clearly state the goal of providing a solid proof for something that has been known for quite a while.

22. Thanks for the constructive comments. As the reviewer has mentioned, the connection between aerobic methane (CH₄) oxidation and denitrification has indeed been previously known, particularly in some specific settings such as bioreactors or pure culture experiments²⁵⁻²⁷. However, empirical evidence on the ubiquity of this connection in complex soil systems is still lacking. Our comprehensive field survey across the transect of flooded croplands in China presents a unique opportunity to address this gap and provide substantial empirical evidence regarding these connections. Additionally, the specific species involved and the associated metabolic pathways responsible for the biochemical interactions within complex flooded croplands remain elusive.

Although our investigation did not unveil new metabolic pathways compared to pure culture or bioreactor experiments, we demonstrated the coupling of the two processes also existed in socially and economically important agricultural croplands and elucidated the underlying mechanisms through a series of field investigations and laboratory experiments. Our empirical evidence bridges the two fundamental processes together, providing a solid theoretical foundation for the accurate prediction of greenhouse gas emissions and nitrogen use efficiency in rice fields. We have revised the manuscript to clearly emphasize our objectives of this study, and provide substantial evidence on the coupling pattern and the underlying mechanisms (species and metabolic interaction) involved in aerobic CH₄ oxidation and denitrification in flooded croplands following the reviewer's comment. Please see our revisions in lines 126-129, 169-170, 192-251, 254-271, 340-390, 427-447, 517-524, 561-564 and 626-627.

I suggest that you expand the Results section to state the goals you tried to pursue with every experiment you conducted. How each experiment leads to the next experiment.

23. That is an important point. We have added texts to state the purpose of each experiment as suggested (Lines 126-129, 169-170 and 192-251). Additionally, we provided a workflow of the conducted experiments in Fig. S6 and expanded the Results section.

In Discussion, provide a brief summary of what is new that you discovered. In my opinion, you present a bigger study on what has been known, but it is important.

24. Thanks for your constructive comments. We have now provided a summary of the study and revised the discussion section to focus on the species and metabolic coupling between CH₄ oxidation and denitrification. Specifically, we summarized that “Our study provides novel insights into the coupling between microbial aerobic CH₄ oxidation and denitrification in flooded croplands harboring diverse aerobic and anaerobic taxa, demonstrating ubiquitous linkages of these two fundamental processes through species and metabolic couplings in soils in various climatic regions.....”. Please see our revisions in lines 254-271.

Please pay more attention to figure legends and expand. For example, what are the symbols a, b, ab etc. stand for?

25. We have carefully reviewed all the figure legends to ensure that no crucial information has been overlooked or omitted. The symbols a, b, c indicate significant differences in treatments using One-way analysis of variance (ANOVA). We have added this information into the figure legends. Thanks a lot for this detailed comment.

What is the role of MDO in methylotrophy? Please explain what it does in the process that you suggest.

26. The *mdo* gene is involved in the oxidation of methanol to formaldehyde by encoding methanol oxidoreductase^{28,29}. The *mdo* gene was enriched in the ¹³C-heavy fraction as indicated by the results of ¹³C-DNA-stable isotope probing (SIP)-metagenomic analysis (Fig. S13), indicating the potential for the production of formaldehyde from methanol during CH₄ oxidation. Further analysis of metagenome-assembled genomes (MAGs) showed that denitrifiers, such as Species Genome Bins (SGBs) belonging to the genus *Novimethylophilus*, possessed the metabolic capacity to utilize formaldehyde, as evidenced by the enrichment of *fae* gene. Therefore, the enrichment of *mdo* gene may indirectly facilitate the coupling of aerobic CH₄ oxidation with denitrification process by regulating the production of formaldehyde. We have incorporated these statements in lines 203-207 and 340-344.

Can you be more specific about which organics might be driving the denitrification process. You mention multicarbon substrates in some instances and monocarbon substrates in others.

27. Our additional ^{13}C -metabolomics experiments reveal that short-chain fatty acids (e.g., acetate, propionate and butyrate) and organic acids associated with pyruvate metabolism and TCA cycle (e.g., lactate) could serve as the C sources for denitrification (Fig. 5B; Figs. 15-16). Specifically, the simultaneous addition of $^{13}\text{CH}_4$ and nitrate (NO_3^-) resulted in an overall increase in the ^{13}C -labeled fraction of acetate, propionate, butyrate and lactate, accompanied by overall decreases in the concentration of these intermediates in both black and yellow soils. These results suggest an increased consumption of $^{13}\text{CH}_4$ -derived intermediates in response to the stimulation of denitrification activities under NO_3^- addition. Additionally, our metagenome-assembled genomes reconstructed from $^{13}\text{CH}_4$ -SIP-metagenomics further revealed the significant enrichment of genes associated with acetate (*acs*), propionate (*pct*), butyrate (*ato*), lactate (*ldh*), malate (*fum*), and succinate (*suc*) utilization within ^{13}C -enriched denitrifiers (Fig. 5A; Fig. 14). As none of these denitrifiers were annotated for the potential of direct CH_4 utilization, the ^{13}C -enriched denitrifiers likely relied on $^{13}\text{CH}_4$ -driven intermediates from methanotrophs. We have incorporated these specific statements in the revised manuscript (Lines 192-252 and 340-384).

How would Rhizobia use methanol? If they are methylotropic do you have proof for that?

28. That is an important point. Although several studies have demonstrated that some Rhizobia species such as those affiliated with genera *Bradyrhizobium* and *Mesorhizobium*, can utilize methanol, the mechanisms through which these strains utilize methanol remain unclear^{30,31}. In this study, the results from $^{13}\text{CH}_4$ -SIP indicate that Rhizobia, as carriers of the denitrification gene *nirK*, have the potential to utilize C from CH_4 . However, further studies are needed to elucidate the specific metabolic pathways employed by these Rhizobia in utilizing the intermediates released during CH_4 oxidation, including methanol. Therefore, additional studies based on pure culture would consolidate our findings in this study. We have carefully discussed this point in the revised manuscript in lines 384-390. Thanks again for all the constructive comments, which have greatly improved the quality of the manuscript.

References

1. Kits, K. D., Klotz, M. G. & Stein, L. Y., Methane oxidation coupled to nitrate reduction under hypoxia by the gammaproteobacterium methylomonas denitrificans, sp. Nov. Type strain FJG1. *Environ. Microbiol.* **17**, 3219-3232 (2015).
2. Chang, J. et al., Enhancement of nitrous oxide emissions in soil microbial consortia via copper competition between proteobacterial methanotrophs and denitrifiers. *Appl. Environ. Microbiol.* **87**, 2301-2320 (2020).
3. Knowles, R., Denitrifiers associated with methanotrophs and their potential impact on the nitrogen cycle. *Ecol. Eng.* **24**, 441-446 (2005).
4. Stein, L. Y. & Klotz, M. G., Nitrifying and denitrifying pathways of methanotrophic bacteria. *Biochem. Soc. Trans.* **39**, 1826-1831 (2011).
5. Hao, Q. et al., Methylobacter couples methane oxidation and N₂O production in hypoxic wetland soil. *Soil. Biol. Biochem.* **175**, 108863 (2022).
6. Turetsky, M. R. et al., A synthesis of methane emissions from 71 northern, temperate, and subtropical wetlands. *Glob. Chang. Biol.* **20**, 2183-2197 (2014).
7. Angle, J. C. et al., Methanogenesis in oxygenated soils is a substantial fraction of wetland methane emissions. *Nat. Commun.* **8**, 1567 (2017).
8. Shi, L.-D. et al., Coupled anaerobic methane oxidation and reductive arsenic mobilization in wetland soils. *Nat. Geosci.* **13**, 799-805 (2020).
9. Lide, D. R. & Frederikse, H., Crc handbook of chemistry and physics, crc press. Inc, Boca Raton, FL, (1995).
10. He, R. et al., Diversity of active aerobic methanotrophs along depth profiles of arctic and subarctic lake water column and sediments. *ISME. J.* **6**, 1937-1948 (2012).
11. Lee, S. et al., Methane-derived carbon flows into host-virus networks at different trophic levels in soil. *Proc. Natl. Acad. Sci. U. S. A.* **118**, e2105124118 (2021).
12. Wigley, K. et al., RNA stable isotope probing and high-throughput sequencing to identify active microbial community members in a methane-driven denitrifying biofilm. *J. Appl. Microbiol.* **132**, 1526-1542 (2022).
13. Huang, T. et al., Ammonia-oxidation as an engine to generate nitrous oxide in an intensively managed calcareous fluvo-aquic soil. *Sci. Rep.* **4**, 3950 (2014).
14. Florio, A., Maienza, A., Dell'Abate, M. T., Stazi, S. R. & Benedetti, A., Changes in the activity and abundance of the soil microbial community in response to the nitrification inhibitor 3,4-dimethylpyrazole phosphate (DMPP). *J. Soils. Sediments.* **16**, 2687-2697 (2016).

15. Bozal-Leorri, A., González-Murua, C., Marino, D., Aparicio-Tejo, P. M. & Corrochano-Monsalve, M., Assessing the efficiency of dimethylpyrazole-based nitrification inhibitors under elevated CO₂ conditions. *Geoderma* **400**, 115160 (2021).
16. Shi, X. et al., Nitrifier-induced denitrification is an important source of soil nitrous oxide and can be inhibited by a nitrification inhibitor 3,4-dimethylpyrazole phosphate. *Environ. Microbiol.* **19**, 4851-4865 (2017).
17. Menéndez, S., Barrena, I., Setien, I., González-Murua, C. & Estavillo, J. M., Efficiency of nitrification inhibitor dmpp to reduce nitrous oxide emissions under different temperature and moisture conditions. *Soil. Biol. Biochem.* **53**, 82-89 (2012).
18. Di, H. J., Cameron, K. C., Podolyan, A. & Robinson, A., Effect of soil moisture status and a nitrification inhibitor, dicyandiamide, on ammonia oxidizer and denitrifier growth and nitrous oxide emissions in a grassland soil. *Soil. Biol. Biochem.* **73**, 59-68 (2014).
19. Conrad, R. Microbial ecology of methanogens and methanotrophs. *Adv. Agron.* **96**, 1-63 (2007).
20. Wang, J. et al., Denitrifying anaerobic methane oxidation: A previously overlooked methane sink in intertidal zone. *Environ. Sci. Technol.* **53**, 203-212 (2019).
21. Roland, F. A. E., Darchambeau, F., Morana, C., Bouillon, S. & Borges, A. V., Emission and oxidation of methane in a meromictic, eutrophic and temperate lake (dendre, belgium). *Chemosphere* **168**, 756-764 (2017).
22. Gardner, L. M. & White, J. R., Denitrification enzyme activity as an indicator of nitrate movement through a diversion wetland. *Soil. Sci. Soc. Am. J* **74**, 1037-1047 (2010).
23. Conrad, R., Frenzel, P. & Cohen, Y., Methane emission from hypersaline microbial mats: Lack of aerobic methane oxidation activity. *FEMS. Microbiol. Ecol.* **16**, 297-305 (1995).
24. Börjesson, G., Sundh, I., Tunlid, A., Frostegård, Å. & Svensson, B. H., Microbial oxidation of CH₄ at high partial pressures in an organic landfill cover soil under different moisture regimes. *FEMS. Microbiol. Ecol.* **26**, 207-217 (1998).
25. Lu, J. J. et al., Simultaneous enhancement of nitrate removal flux and methane utilization efficiency in MBFR for aerobic methane oxidation coupled to denitrification by using an innovative scalable double-layer membrane. *Water. Res.* **194**, 116936 (2021).
26. Costa, C. et al., Denitrification with methane as electron donor in oxygen-limited bioreactors. *Appl. Microbiol. Biot.* **53**, 754-762 (2000).

27. Kalyuzhnaya, M. G. et al., Highly efficient methane biocatalysis revealed in a methanotrophic bacterium. *Nat. Commun.* **4**, 2785 (2013).
28. Kolb, S. & Stacheter, A., Prerequisites for amplicon pyrosequencing of microbial methanol utilizers in the environment. *Front. Microbiol.* **4**, 268 (2013).
29. Park, H., Lee, H., Ro, Y. T. & Kim, Y. M., Identification and functional characterization of a gene for the methanol : N,N'-dimethyl-4-nitrosoaniline oxidoreductase from Mycobacterium sp. Strain JC1 (DSM 3803). *Microbiology* **156**, 463-471 (2010).
30. Jaftha, J. B., Strijdom, B. W. & Steyn, P. L., Characterization of pigmented methylotrophic bacteria which nodulate lotononis bainesii. *Syst. Appl. Microbiol.* **25**, 440-449 (2002).
31. Kanukollu, S. et al., Methanol utilizers of the rhizosphere and phyllosphere of a common grass and forb host species. *Environ. Microbiome.* **17**, 35 (2022).

Reviewers' Comments:

Reviewer #1:

Remarks to the Author:

The authors addressed all my concerns and I can only recommend this manuscript for publication.

Reviewer #2:

Remarks to the Author:

Manuscript Number: NCOMMS-23-14274A

Title: Species interactions and metabolic couplings between soil aerobic methanotrophs and denitrifiers in flooded croplands

The author carefully revised the MS and added a large amount of data to reveal the coupling of aerobic CH₄ oxidation and denitrification in flooded farmland. The author also added new methods, including MAGs and ¹³C metabolomics. After careful consideration, I think the novelty of this MS has improved compared to the previous version. I am certain of the author's efforts in terms of research breadth and data volume, but I still have some confusions about the mechanism and couplings of the two processes. Therefore, I suggest the author to further explore the following issues.

1. Just as one of the reviewers pointed out, although the authors provided detailed information on metabolic processes and intermediate product determination, they did not discover any new metabolic pathways between the two processes mentioned above. I found that the author used ¹³C-MAGs to provide intermediate products of methane oxidation as substrates for denitrification. However, short chain carbon could be served as substrates for denitrification, which has been extensively studied in reactors. So, what are the differences of the denitrification process driven by short chain carbon between reactors and paddy soil? Or what is the significance of these potential carbon sources for denitrification processes?

2. In fact, I believe that the author's newly added methods (currently some of the more common methods) have not fundamentally improved the innovation of this article, but have only enriched more details and process information. I acknowledge that the author's research has revealed the interrelationship between aerobic oxidation and denitrification of methane in rice fields and provided more details. However, what is the connection between these studies and NUE? I found no relevant data in the MS to support the logical relationship between the two. (Line 270-271)

3. The author used ¹⁵N isotopes to determine the rate of denitrification and claim that denitrification activity did not alter the overall patterns across different climatic regions, nor did it affect the relationships with CH₄ oxidation activity. Does denitrification rate not be affected by methane oxidation activity? The added data demonstrated that the intermediates of methane oxidation can be served as substrates for denitrification. So, the activity of methane oxidation means that the concentration of different intermediate products varies, and the denitrification rate should change. Why does the author believe that denitrification activity and methane oxidation activity do not affect each other? The author should provide more details on the effects of denitrification rate on metabolic couplings.

Reviewer #4:

Remarks to the Author:

My focus was mainly on the analysis of the MAGs. However, I have also included some technical comments not focusing on the MAGs.

The approach for MAG binning is solid although some details are missing. First, did you do individual assemblies or co-assembled all reads? Were assemblies conducted with the default settings? How about Metabat2, what was the length cut-off for assembled contigs? The number of MAGs is rather low so I would like to see some assembly statistics. What was the length of the assembly/assemblies and how many contigs? With coverM, how much of the reads were mapped back to the assemblies?

What was the KEGG database version used? For the denitrification genes misannotation can occur with blast based approaches. I would advice you to check the ORFs for the presence of conserved residues at positions associated with the binding of co-factors and active sites. And use HMM based serach tool against some more specific database (eg. <https://doi.org/10.1371/journal.pone.0114118>).

You mention novel species (L 197), but what does this mean? Did you run ANI or some other measures to calculate that these 7 identified species are novel species? Please add this information (ANI values or similar) and how this was measured. I did not see any report on mapping RNA transcripts to the MAGs. Were the enriched MAGs active in the samples? Activity of those linked N and C cycling genes would give support to the findings.

For me figure 5 is not that clear. You aim to show the differences in metabolic pathways of six MAGs in different soils. Having all six MAGs in one figure makes it rather busy and to make it more clear without making it too large, would have 2 metabolic figures next to each make it more clear. For example the methanotrophic MAGs in the left had side and denitrifiers on the right hand side? As the idea is to emphasize both processed within these MAGs? What was the rationale for log (n+1) normalization? I considerer for example TPM normalization better as the varying gene length are taken into account. The rather small shapes with count information are also difficult to differentiate.

Last, in the discussion you mention Species Genome Bins. Why not MAGs? This should be elaborated a bit more, why highlight SGBs?

- Line 510-512, please add references to the tools used and SILVA database version to line 515
- Line 541, to my knowledge RNA quality cannot be measured with Nanodrop. Also was the potential DNA contamination in RNA fractions measured with PCR? This should be verified to really analyze RNA transcripts and not DNA contamination.
- line 577 How DNA was extracted from the SIP experiments? How much soil in extraction, which method to purify? Were replicate extractions done?
- Line 596, what was the rationale for using 85% similarity for these genes?
- Line 603 which kit was used for paired-end libraries (please collect the typo)?
- Line 608, perhaps having the exact read number is not needed but the numbers could be rounded up to closest Mb
- L 264: as you had metabolites this is more than just a potential.

Response letter

Responses to reviewer's comments:

Reviewer #1:

The authors addressed all my concerns and I can only recommend this manuscript for publication.

1. Thanks for your positive assessments and all the efforts in reviewing our manuscript.

Reviewer #2:

The author carefully revised the MS and added a large amount of data to reveal the coupling of aerobic CH₄ oxidation and denitrification in flooded farmland. The author also added new methods, including MAGs and ¹³C metabolomics. After careful consideration, I think the novelty of this MS has improved compared to the previous version. I am certain of the author's efforts in terms of research breadth and data volume, but I still have some confusions about the mechanism and couplings of the two processes. Therefore, I suggest the author to further explore the following issues.

2. Thank you for your comprehensive review and the positive comments on our efforts in the research breadth and data volume of the manuscript. We appreciate your constructive suggestions regarding the coupling mechanisms of methane (CH₄) oxidation and denitrification. We have carefully considered your feedback and made substantial revisions to enhance the clarity of these mechanisms. In the revised manuscript, we have thoroughly explained our results to elucidate the coupling mechanisms. We now provide our point-to-point responses to your concerns as follows.

1. Just as one of the reviewers pointed out, although the authors provided detailed information on metabolic processes and intermediate product determination, they did not discover any new metabolic pathways between the two processes mentioned above. I found that the author used ¹³C-MAGs to provide intermediate products of methane oxidation as substrates for denitrification. However, short chain carbon could be served as substrates for denitrification, which has been extensively studied in reactors. So, what are the differences of the denitrification process driven by short chain carbon between reactors and paddy soil? Or what is the significance of these potential carbon sources for denitrification processes?

3. Thank you for your constructive comments and recognition of the advancements in our manuscript. We acknowledge your point about the absence of newly identified metabolic pathways in comparison to bioreactor studies. However, the identification of novel metabolic pathways predominantly relies on methodologies involving pure cultures, which poses significant challenges in soil systems. The inherent challenges stem from the volatile and unstable nature of metabolites, which transform rapidly and have intricate interplay with different soil components, such

as minerals and organic matter. Nevertheless, our study identified key microbial groups involved in aerobic CH₄ oxidation and denitrification processes using ¹³CH₄-metagenome-assembled genomes (MAGs) and ¹³C-DNA-stable isotope probing (SIP) methods. This could be pivotal in pinpointing relevant strains and pure cultures, thereby laying a robust foundation for future explorations into novel metabolic pathways.

Indeed, as you highlighted, the novelty of our work lies in investigating the relative importance of different carbon sources for denitrification processes, which remained ambiguous due to the lack of direct measurements of ¹³C-metabolomics within complex soil systems. While ¹³C-metabolomics has been employed in pure culture studies to identify CH₄ oxidation metabolites, its application in the multifaceted environment of soil ecosystems posed significant challenges. Previous bioreactor-based studies primarily focused on quantifying the variations in the concentration of some CH₄ oxidation intermediates or employing ¹³C-labeled small molecules (e.g., ¹³C-methanol, ¹³C-acetate) to investigate their role in promoting denitrification. Our study extends beyond the traditional scope by identifying a comprehensive range of intermediates of CH₄ oxidation by directly measuring ¹³C-labeled metabolites. Moreover, we dissected the relative significance of different metabolic pathways by comparing the concentrations and ¹³C-signals of various metabolites across different types of soils. Undertaking this comprehensive approach in the inherently intricate soil ecosystems presented substantial methodological challenges, necessitating significant efforts in the isolation and purification of ¹³C metabolites within the complex soil systems, as well as in the subsequent data analysis. We have clarified the coupling mechanism between CH₄ oxidation and denitrification in lines 346-371.

2. In fact, I believe that the author's newly added methods (currently some of the more common methods) have not fundamentally improved the innovation of this article, but have only enriched more details and process information. I acknowledge that the author's research has revealed the interrelationship between aerobic oxidation and denitrification of methane in rice fields and provided more details. However, what is the connection between these studies and NUE? I found no relevant data in the MS to support the logical relationship between the two. (Line 270-271)

4. Thanks for your valuable feedback. We appreciate your recognition of the detailed process information added to the manuscript. As responded above, the novelty of our research lies in the direct comparative analysis of the full range of CH₄ oxidation metabolites and the evaluation of their relative importance in the coupling between CH₄ oxidation and denitrification within the complex rice-field ecosystems. We acknowledge the earlier oversight in sufficiently supporting the connection between our findings and N fertilizer use efficiency (NUE). To maintain the integrity and accuracy of our research, we have removed the statements regarding NUE in the discussion section that may have overextended the interpretation of our results. This revision ensures that our conclusions are firmly supported by the data presented.

3. The author used ^{15}N isotopes to determine the rate of denitrification and claim that denitrification activity did not alter the overall patterns across different climatic regions, nor did it affect the relationships with CH_4 oxidation activity. Does denitrification rate not be affected by methane oxidation activity? The added data demonstrated that the intermediates of methane oxidation can be served as substrates for denitrification. So, the activity of methane oxidation means that the concentration of different intermediate products varies, and the denitrification rate should change. Why does the author believe that denitrification activity and methane oxidation activity do not affect each other? The author should provide more details on the effects of denitrification rate on metabolic couplings.

5. Thanks for highlighting the need for clarity in our manuscript. Our intention was to convey that the results obtained using ^{15}N isotopes for denitrification rates were consistent with those measured using nitrous oxide (N_2O) production. This consistency underpins our claim that the utilization of ^{15}N isotopes does not significantly alter the overarching patterns observed across different climatic regions, nor does it impact its relationship with CH_4 oxidation activity, when compared with the measurements derived from N_2O production.

Indeed, both ^{15}N isotopes and N_2O production methods revealed that denitrification activity varied across different climatic zones, as illustrated in Fig. 1 and detailed in lines 107-109 of the manuscript. These results underscore the environmental heterogeneity and complexity of denitrification processes. Following your suggestions in the previous review, we used ^{15}N isotope methods to improve the assessment of denitrification activity. Moreover, a key finding of our study is the observed positive correlation between CH_4 oxidation and denitrification, indicating a potential linkage between these two processes in flooded croplands across the national scale. To further substantiate this notion, we conducted a series of microcosm experiments, including ^{13}C - CH_4 -DNA-SIP, CH_4 and methanotroph addition, and ^{13}C -metabolomics analyses. These additional experiments provide more solid evidence demonstrating metabolic interconnections between CH_4 oxidation and denitrification, reinforcing our initial large-scale observations. These findings and their implications have been thoroughly discussed in the manuscript, particularly in lines 328-341 and 342-397.

Reviewer #4:

My focus was mainly on the analysis of the MAGs. However, I have also included some technical comments not focusing on the MAGs.

6. Thanks for all your constructive comments. We have carefully addressed all your concerns about the technical methods. In particular, we have incorporated detailed information in the methods section including the analyses of metagenome-assembled genomes (MAGs) and amplicon sequencing. Please see our point-to-point responses below.

The approach for MAG binning is solid although some details are missing. First, did you do individual assemblies or co-assembled all reads? Were assemblies conducted with the default settings? How about Metabat2, what was the length cut-off for assembled contigs? The number of MAGs is rather low so I would like to see some assembly statistics. What was the length of the assembly/assemblies and how many contigs? With coverM, how much of the reads were mapped back to the assemblies?

7. Thanks for these insightful comments. In response to your queries, we conducted individual assemblies for the qualified reads using Megahit (<https://github.com/voutcn/megahit>, -k-min 47 -k-max 97 -k-step 10). Subsequently, Metabat2 was used to bin the contigs assembled from three replicates, specifically targeting those with lengths exceeding 1000 bp. The outcomes of this process yielded 85 MAGs in the black soil, 69 MAGs in the red soil, and 100 MAGs in the yellow soil. The assembly lengths for the assembly statistics were 444.10-811.21 Mb, comprising 621075-1206876 contigs in total. Furthermore, we employed CoverM (v0.6.1) to estimate the coverage of MAGs (%) in each sample. The results indicated that 21.28% to 49.07% of the reads were successfully mapped back to the assemblies. To enhance the clarity of our manuscript, we have provided additional details in lines 636-644 and 647-649 in the revised manuscript.

What was the KEGG database version used? For the denitrification genes misannotation can occur with blast based approaches. I would advice you to check the ORFs for the presence of conserved residues at positions associated with the binding of co-factors and active sites. And use HMM based search tool against some more specific database (eg. <https://doi.org/10.1371/journal.pone.0114118>).

8. The version of the KEGG database for functional annotation was KEGG v 202209. We have carefully checked the ORFs associated with the denitrification genes following your suggestion to avoid any misannotations with blast-based approaches. Furthermore, we have re-annotated genes within MAGs using the HMM profile database for KEGG orthology via KofamScan (v 1.3.0)¹. Specifically, for denitrification genes, we employed HMM-based search tools against the curated database referenced in the article you have recommended². Our analysis revealed that only a few denitrification genes were misannotated, and we have corrected these discrepancies in both Fig. 5 and S14. For further details, please see our corresponding revisions in lines 198-227 and 656-661.

You mention novel species (L 197), but what does this mean? Did you run ANI or some other measures to calculate that these 7 identified species are novel species? Please add this information (ANI values or similar) and how this was measured. I did not see any report on mapping RNA transcripts to the MAGs. Were the enriched MAGs active in the samples? Activity of those linked N and C cycling genes would give support to the findings.

9. The term “novel species” refers to bacterial species that have not been previously documented or classified. In this study, ANI analysis (with a threshold of 95%) was conducted, revealing that all MAGs meet the criteria for novel species³. This

information has been clarified in both the methods (lines 652-653) and the results section (lines 194-198).

Given the novelty of these species, they were no correspondence with any species in the RNA transcript data. Despite the absence of transcript data, it is important to note that the MAGs were assembled from ^{13}C -metagenomes. The combination of stable isotope probing (SIP) with metagenomes has demonstrated efficacy in uncovering valuable insights into active microbes for specific functions such as C and N cycles. Further studies on these potential novel species will contribute to reinforcing the robustness of our results obtained from the analysis of ^{13}C -MAGs.

For me figure 5 is not that clear. You aim to show the differences in metabolic pathways of six MAGs in different soils. Having all six MAGs in one figure makes it rather busy and to make it more clear without making it too large, would have 2 metabolic figures next to each make it more clear. For example the first MAGs in the left had side and secend on the right hand side? As the idea is to emphasize both processed within these MAGs? What was the rationale for $\log(n+1)$ normalization? I considerer for example TPM normalization better as the varying gene length are taken into account. The rather small shapes with count information are also difficult to differentiate.

10. We appreciate your constructive comments. In response to your valuable suggestion, we have made revisions to Fig. 5 to improve clarity. The updated figure now represents the common and unique metabolic pathways of six MAGs. The left side of Fig. 5 specifically highlights CH_4 oxidation, while the right side emphasizes denitrification processes. Additionally, we have calculated Transcripts Per Million (TPM), taking into account the lengths of varying genes, to better reflect the relative abundance. The $\log(\text{TPM}+1)$ was visualized in Fig. 5 following previous studies⁴⁻⁶. The $\log(n+1)$ transformation compressed the data scale and spread out values that were clustered towards lower values while preserving relative differences. Adding 1 before taking the logarithm ensured the transformation was defined for zero values. Furthermore, we have updated the color gradient to three distinct levels in Fig. 5, facilitating a clearer distinction in count information among MAGs. Please see our revisions in Fig. 5 and lines 660-662.

Fig. 5 The proposed metabolic pathways of the coupling between aerobic methane (CH_4) oxidation and denitrification in the flooded soils. A The color gradients represent the logarithm of transcripts per million (TPM) of major genes in the corresponding metagenome-assembled genomes (MAGs) classified as methanotrophs and denitrifiers, respectively, in the heavy DNA from $^{13}CH_4$ incubation. The definitions of genes are listed in Table S7. B Changes in the ^{13}C -labeled fraction of metabolites derived from $^{13}CH_4$ oxidation in the three typical soils. The error bar represents the standard error of triplicate samples ($n = 3$). Different lowercase letters in B indicate significant differences between treatments ($p < 0.05$).

Last, in the discussion you mention Species Genome Bins. Why not MAGs? This should be elaborated a bit more, why highlight SGBs?

11. We appreciate your insightful comment. Upon dereplicating our database and aligning our sequences with known species in the Genome Taxonomy Database (GTDB) at a 95% ANI threshold, we reclassified our MAGs as Species Genome Bins (SGBs). This terminology is in line with a standard species-level delineation⁷. We referenced SGBs in our study to bring out the phylogenetic insights of the MAGs, following numerous previous studies⁷⁻⁹.

Line 510-512, please add references to the tools used and SILVA database version to line 515

12. Added as suggested in lines 519-522.

Line 541, to my knowledge RNA quality cannot be measured with Nanodrop. Also was the potential DNA contamination in RNA fractions measured with PCR? This should be verified to really analyze RNA transcripts and not DNA contamination.

13. We understand the reviewer's concern. Following the extraction of total RNA from all samples, we conducted an assessment of RNA quality through agarose gel electrophoresis. Notably, clear bands corresponding to 28S, 18S, and 5S were observed in all samples, confirming the high quality of the total RNA. Subsequently, cDNAs were synthesized using the PrimeScriptTM RT reagent kit with gDNA Eraser (Takara) following the manufacturer's protocol. To prevent contamination with DNA, RNA fraction was applied to DNase treatment, and 2 μ L of RNA was extracted as contamination control after DNase treatment and before cDNA synthesis. The extracted RNA was then subjected to PCR amplification to check for the presence of any DNA contamination. Importantly, no PCR products were observed in any of the RNA samples after DNase treatment, providing solid evidence for the absence of DNA contamination in RNA fractions. We have supplemented these details in lines 548-559.

Line 577 How DNA was extracted from the SIP experiments? How much soil in extraction, which method to purify? Were replicate extractions done?

14. Microbial DNA was extracted from 0.25 g freeze-dried soil using the MoBio PowerSoil DNA Isolation Kit (MoBio Laboratories, Carlsbad, CA, USA) following the instructions. The DNA extraction procedure with this kit included mechanical and chemical lysis of cells and subsequent column-based DNA purification¹⁰. To ensure the reliability of our results, we performed three independent biological replicates including the entire process of DNA extraction from soil. This approach not only provided statistical confidence in our findings but also ensured the reproducibility of our results across different soil samples. We have included the information in lines 590-595.

Line 596, what was the rationale for using 85% similarity for these genes?

15. We acknowledge the reviewer's concern regarding the rationale for selecting 85% similarity threshold for clustering gene sequences. The selection of a similarity threshold is a critical decision that necessitates a careful balance to prevent both the

excessive aggregation of functionally divergent sequences and the insufficient clustering of sequences that are functionally similar. Previous research has established that a similarity threshold within the range of 80-90% effectively balances biological diversity and conservatism for genes involved in CH₄ oxidation and denitrification processes^{11, 12}. In our study, we aimed to determine the most suitable similarity threshold for OTU clustering by methodically performing clustering at various thresholds, spanning from 60% to 97%. Notably, at the 85% similarity threshold, we identified an inflection point in the OTU count (see Fig. S20). This threshold was selected as it best met our analytical needs, taking into account the specific features of the soil ecosystem and the functional genes under investigation. Therefore, we used 85% threshold following the previous study¹¹. We have incorporated these details into the manuscript in lines 613-624.

Line 603 which kit was used for paired-end libraries (please collect the typo)?

16. The paired-end library was constructed using NEXTFLEX Rapid DNA-Seq (Bioo Scientific, Austin, TX, USA). We have corrected the typo and supplemented this detail in lines 631-633.

Line 608, perhaps having the exact read number is not needed but the numbers could be rounded up to closest Mb.

17. We have revised the sentences as suggested in lines 636-637 “As a result, a total of 10162.46-18720.82 Mb (10.16-18.72 Gb) clean reads were kept, accounting for 92.6-97.8% of raw bases.”

L 264: as you had metabolites this is more than just a potential.

18. Thanks for your constructive comments. We have rewritten this sentence in lines 264-267 as “Our ¹³C-MAGs and ¹³C-metabolomics measurements further indicate that denitrifiers could utilize intermediate compounds derived from aerobic CH₄ oxidation, such as acetate, propionate, butyrate and lactate, thereby enabling the coupling between aerobic CH₄ oxidation and denitrification.”

References

1. Aramaki, T. et al., Kofamkoala: KEGG ortholog assignment based on profile hmm and adaptive score threshold. *Bioinformatics* **36**, 2251-2252 (2020).
2. de Crécy-Lagard, V., Graf, D. R. H., Jones, C. M. & Hallin, S., Intergenomic comparisons highlight modularity of the denitrification pathway and underpin the importance of community structure for N₂O emissions. *PLoS. One.* **9**, e114118 (2014).
3. Jain, C., Rodriguez, R. L., Phillippy, A. M., Konstantinidis, K. T. & Aluru, S., High throughput ani analysis of 90k prokaryotic genomes reveals clear species boundaries. *Nat. Commun.* **9**, 5114 (2018).
4. Arora, J. et al., The functional evolution of termite gut microbiota. *Microbiome* **10**, 78 (2022).
5. Fan, L. et al., Presence and role of viruses in anaerobic digestion of food waste under environmental variability. *Microbiome* **11**, 170 (2023).
6. Martinez-Perez, C. et al., Phylogenetically and functionally diverse microorganisms reside under the ross ice shelf. *Nat. Commun.* **13**, 117 (2022).
7. Ma, B. et al., A genomic catalogue of soil microbiomes boosts mining of biodiversity and genetic resources. *Nat. Commun.* **14**, 7318 (2023).
8. Ke, S., Weiss, S. T. & Liu, Y.-Y., Dissecting the role of the human microbiome in COVID-19 via metagenome-assembled genomes. *Nat. Commun.* **13**, 5235 (2022).
9. Chen, C. et al., Expanded catalog of microbial genes and metagenome-assembled genomes from the pig gut microbiome. *Nat. Commun.* **12**, 1106 (2021).
10. Videvall, E., Strandh, M., Engelbrecht, A., Cloete, S. & Cornwallis, C. K., Direct pcr offers a fast and reliable alternative to conventional DNA isolation methods for gut microbiomes. *MSystems* **2**, e00132-00117 (2017).
11. Murdock, S. A. & Juniper, S. K., Capturing compositional variation in denitrifying communities: A multiple-primer approach that includes epsilonproteobacteria. *Appl. Environ. Microbiol.* **83**, e02753-02816 (2017).
12. Li, C. et al., Global co-occurrence of methanogenic archaea and methanotrophic bacteria in microcystis aggregates. *Environ. Microbiol.* **23**, 6503-6519 (2021).

Reviewers' Comments:

Reviewer #2:

Remarks to the Author:

The authors addressed all my concerns and now I recommend this manuscript for publication.

Reviewer #5:

Remarks to the Author:

The authors' responses to reviewer #4 suggestions are well-conceived and appropriately address any concerns previously raised. With that said, I strongly encourage the authors to classify bin quality (high-, medium-, or low-quality) in accordance with MIMAG standards (<https://www.nature.com/articles/nbt.3893>), which goes beyond completeness and contamination scores.

Response letter

Responses to reviewer's comments:

Reviewer #2:

The authors addressed all my concerns and now I recommend this manuscript for publication.

1. We're glad that the reviewer is satisfied with our revising. Thanks a lot for the positive feedbacks and all the efforts in reviewing our manuscript.

Reviewer #5:

The authors' responses to reviewer #4 suggestions are well-conceived and appropriately address any concerns previously raised. With that said, I strongly encourage the authors to classify bin quality (high-, medium-, or low-quality) in accordance with MIMAG standards (<https://www.nature.com/articles/nbt.3893>), which goes beyond completeness and contamination scores.

2. We appreciate the positive comments and satisfaction in our responses to the previous concerns. We greatly value the suggestions regarding the classification of genome bin quality according to MIMAG standards (Bowers et al., 2017, Nat Biotechnol). We thus have re-classified genome bins with both medium- and high-quality according to this reviewer's suggestion, by considering completeness, contamination, and the presence of genes at specific locations, such as 23S, 16S, 5S rRNA and at least 18 tRNAs. Please see our classification in Supplementary Table 2.

Specifically, we identified 30 MAGs meeting the medium-quality criteria affiliated with methanotrophs and denitrifiers, though few matched the high-quality criteria. This absence of high-quality MAGs could be due to that the overly stringent standards recommended by MIMAG require long-read sequencing, which could be more efficiently obtained using third-generation sequencing but is currently prohibitive costs (Jin et al., 2023, Nat Microbiol). Importantly, our metagenomic sequencing was conducted on the DNA extracted from ¹³C-labeled samples in the ¹³CH₄-DNA-SIP experiments, which substantially reduced the quantity of extracted DNA by an order of magnitude or more. Given these considerations, we carefully considered MAGs with >80% completeness and <10% contamination as "fairly complete genomes", following many previous studies (Hua et al., 2019, Nat Commun; Zhou et al., 2020, ISME J; Stewart et al., 2019, Nat Biotechnol; Walsh et al., 2020, Nat Food). Moreover, we have toned down our statements to avoid misleading (e.g., lines 198-199, 218 and 674).

References

Bowers, R. M. et al., Minimum information about a single amplified genome (MISAG) and a metagenome-assembled genome (MIMAG) of bacteria and archaea. *Nat. Biotechnol.* **35**, 725-731 (2017).

Jin, H. et al., A high-quality genome compendium of the human gut microbiome of inner mongolians. *Nat. Microbiol.* **8**, 150-161 (2023).

Hua, Z. S. et al., Insights into the ecological roles and evolution of methyl-coenzyme m reductase-containing hot spring archaea. *Nat. Commun.* **10**, 4574 (2019).

Zhou, Z., Tran, P. Q., Kieft, K. & Anantharaman, K., Genome diversification in globally distributed novel marine proteobacteria is linked to environmental adaptation. *ISME. J.* **14**, 2060-2077 (2020).

Stewart, R. D. et al., Compendium of 4,941 rumen metagenome-assembled genomes for rumen microbiome biology and enzyme discovery. *Nat. Biotechnol.* **37**, 953-961 (2019).

Walsh, A. M., Macori, G., Kilcawley, K. N. & Cotter, P. D., Meta-analysis of cheese microbiomes highlights contributions to multiple aspects of quality. *Nat. Food.* **1**, 500-510 (2020).